# Systems Thinking Accident Analysis Models: A Systematic Review for Sustainable Safety Management

Mahdieh Delikhoon [1], Esmaeil Zarei [2], Osiris Valdez Banda [3], Mohammad Faridan [4] and Ehsanollah Habibi [1,*]

1 Department of Occupational Health and Safety Engineering, Faculty of Health,
Isfahan University of Medical Sciences, Isfahan 81746-73461, Iran; mdelikhon@yahoo.com
2 Centre for Risk, Integrity and Safety Engineering (C-RISE), Faculty of Engineering and Applied Science,
Memorial University of Newfoundland, St. John's, NL A1B 3X5, Canada; ezarei@mun.ca
3 Research Group on Maritime Risk and Safety, Department of Applied Mechanics, Aalto University,
00076 Espoo, Finland; osiris.valdez.banda@aalto.fi
4 Environmental Health Research Center, Department of Occupational Health and Safety at Work Engineering,
Lorestan University of Medical Sciences, Khorramabad 68138-33946, Iran; mfereidan@yahoo.com
* Correspondence: habibi@hlth.mui.ac.ir; Tel.: +98-913-113-1623

**Abstract:** Accident models are mental models that make it possible to understand the causality of adverse events. This research was conducted based on five major objectives: (i) to systematically review the relevant literature about AcciMap, STAMP, and FRAM models and synthesize the theoretical and experimental findings, as well as the main research flows; (ii) to examine the standalone and hybrid applications for modeling the leading factors of the accident and the behavior of sociotechnical systems; (iii) to highlight the strengths and weaknesses of exploring the research opportunities; (iv) to describe the safety and accident models in terms of safety-I-II-III; and finally, to investigate the impact of the systemic models' applications in enhancing the system's sustainability. The systematic models can identify contributory factors, functions, and relationships in different system levels which helps to increase the awareness of systems and enhance the sustainability of safety management. Furthermore, their hybrid extensions can significantly overcome the limitations of these models and provide more reliable information. Applying the safety II and III concepts and their approaches in the system can also progress their safety levels. Finally, the ethical control of sophisticated systems suggests that further research utilizing these methodologies should be conducted to enhance system analysis and safety evaluations.

**Keywords:** accident analyses; AcciMap; STAMP; FRAM; safety-III; sustainable system

## 1. Introduction

The protection of human resources and environments along with reducing the risk of losses are the major concerns of system managers all over the world. Safety management has also shown to have a vital role in establishing the sustainable progress of a system [1]. The concept of sustainability refers to the effective management of the environment in short and long-term procurement in order to ensure that resources and social provisions meet the needs of future generations. It also takes into account the potential for long-term risk reduction [2].

In that regard, establishing a sustainable organization requires proactively managing risk in an integrated way to decrease unplanned chains of events and losses—particularly, in order to promote the quality of performance and productivity. On the other hand, one of the key elements for achieving sustainability, improving safety, and maintaining low incident rates is to perform a comprehensive, accurate and detailed analysis of an organization's incidents and accidents [3].

An accident is defined as an unplanned chain of events resulting from inadequate risk control or the application of safety constraints that causes injury, illness or damage to

people, property, the environment, or credit [4]. The ILO states that occupational accidents or illnesses cause the death of one worker every 15 s. It also declares that 153 accidents occur due to work practices at the same time, and 6300 workers die every day from work-related illnesses or accidents at work.

The ILO also declares that shortcomings in taking appropriate health and safety measures at work lead to an economic burden equal to 4% of global GDP per year [5–7]. Illnesses and accidents induced by work activities have also proved to affect economic growth much more than several other common illnesses and disorders, such as cancer, cardiovascular disorders, Alzheimer, and HIV/AIDS [6]. It is worth noting that the socio-economic costs of accidents are significantly higher than their financial ones and such costs cannot be easily estimated. This highlights the importance of risk assessment, reliability analysis and modeling of the causation of the accidents [8,9].

Occupational accidents usually occur due to several factors, such as human factors, job design, environmental and economic conditions, lack of experience, long working hours, fatigue, sleep disorders, noise, physical pressures, workload, role ambiguity and conflicts, and demographic characteristics and lifestyle [10–18].

Some studies suggest that the human factors contribute to approximately 80% of occupational accidents and that human error is a main contributing factor for workplace accidents [13,19].

Most industrial facilities are complex engineered sociotechnical systems where the social, human, organizational, and technical factors are considered in their design and structure. Internal and external interactions between physical equipment and people also exist in such facilities [20]. In other words, with the increasing advancement of technology and complex engineering systems, accidents are not simply the result of a minor failure. Although they emerge from complex interactions between system components, they are usually related to latent factors such as human error, technical failures, external factors and abnormal process situations [21]. Due to the complexity of modern industrial technological systems, the risk of accidents involving such systems has become more concerning [22,23]. The continual recurrence of catastrophic events such as Bhopal, Piper Alpha, BP Texas City, Bunce field, and Gulf of Mexico, as complex technological systems, has contributed to serious losses and raised social and legislative stakeholders' concerns over the last decades [24,25]. The accident in the Gulf of Mexico highlighted some critical issues in system safety and common thinking about defining the causality of accidents. It also revealed that the linear models are incapable of determining the interaction between the leading factors, and, despite their wide use in accident analysis techniques, do not enable systems to reach the zero-accident target [24]. Therefore, as highlighted by Hollnagel et al. (2006), in order to control the adverse consequences of these accidents, it is essential to know the background, future complications, control measures, and resources that can be achieved through using accident modeling strategies [26]. In other words, accident models are mental models upon which it is possible to understand how and why accidents occur in terms of causality. They are also used as a means of risk assessment to determine appropriate safety measures for enhancing the stability of systems [27,28]. Therefore, these concepts have been promoted in recent years as effective tools in enhancing safety and preventing accidents through applying proactive rather than reactive methods. The most important step necessary to achieve this goal is to enhance awareness about the technological, organizational and human factors affecting the system [3].

Various classifications of the accident models have therefore been introduced and evaluated in the literature [29–38]. Accident models are usually divided into sequential, epidemiological and systemic models [39]. While the focus of the first two models is on the linear investigation of accident causality, the systemic models mainly consider the interaction among the major system components (technical, human, organizational, and managerial). In other words, the interrelations among the causes of the accident according to the systematic model are non-linear and include multiple feedback loops [40,41].

Nonetheless, the application of these advanced models and their associated methods have already been expanded and criticized at a number of different levels [42,43]. Therefore, it is timely to systematically subject the studies of accident analysis models to a thorough review. Furthermore, much of the research in this field, up to now has focused on the review of the specific methodologies (e.g., AcciMap) or distinctive accident models [44]. Hence, we believe that broad review on systemic analysis methods should be conducted to fully provide ample indications about how they can be more applicable to conduct practical analysis as well as preventing the accidents.

Therefore, the principal objectives of this systematic review were defined as follows: First, an overview of the papers that had applied the methodologies of AcciMap, STAMP, FRAM in their analyses to synthesize the theoretical and experimental findings—particularly for recognizing the main research flows. Second, to examine the application of the mentioned approaches combined with other methods for modeling causal factors of the accidents and the behavior of sociotechnical systems. Third, highlighting the advantages and disadvantages of these approaches to explore the opportunities for research and practice. Fourth, to describe the safety and accident models in terms of safety-I ("as few things as possible go wrong") and safety-II ("as many things as possible go right"), as well as safety-III ("freedom from unacceptable losses"). To describe these three paradigms of safety in detail: In the safety-I paradigm, accidents occur due to system failures and performance malfunctions, according to which safety management is reactive because the response is to the time that events occurred and any contributory factors were identified. In the safety-II paradigm, the system is adjusted to respond to events and to eliminate the problems before they occurred and its effort is to make functions "go right". Based on this concept, safety management is proactive. The safety-III concept represents that inadequate hazards control is the main cause of accidents. In this paradigm, safety management does not regard the identification of the root cause. Instead, it investigates the reason for control malfunctions, preventing accidents, and system performance auditing [4,45].

The final objective of this work was to investigate the impact of employing the systemic models for enhancing the systems sustainability.

Accordingly, the following research questions were defined for this review:

What research flows in sociotechnical systems have been examined from the perspective of these three systemic accident models?
How has previous research contributed to the three systemic accident models and what are the needs and shortcomings in these studies?
How are the current problems best dealt with and what challenges do accident analysts face?
What is the role of systemic accident models in improving system sustainability?

### 1.1. Evaluation of Accident Models

Generally, there are three categories of accident models: sequential, epidemiological and systemic models [46]. The classification of these models and their subset methods are illustrated in Figure 1.

### 1.2. Sequential Accident Models

According to these models, the leading cause of an accident is a linear sequence of events. In other words, the causes of these accidents stem from a series of separate events that occur in a specific chronological order. Most of the traditional accident models such as Domino theory, CCA, FTA, ETA, and FMEA are classified within this type. Domino theory is different from domino effect as the second involves extensive resonance creating events in the process and chemical industries [39,47]. This category, however, suffers from some limitations in determining the contributing factors of the accidents in the complex sociotechnical systems that were developed in the second half of the twentieth century [48]. Accidents have always proved to have more than just one single cause. Thus, the need for more robust methods of overcoming the limitation of sequential models that explain the

underlying causes of accidents lead to the development of epidemiological models in the 1980s [49].

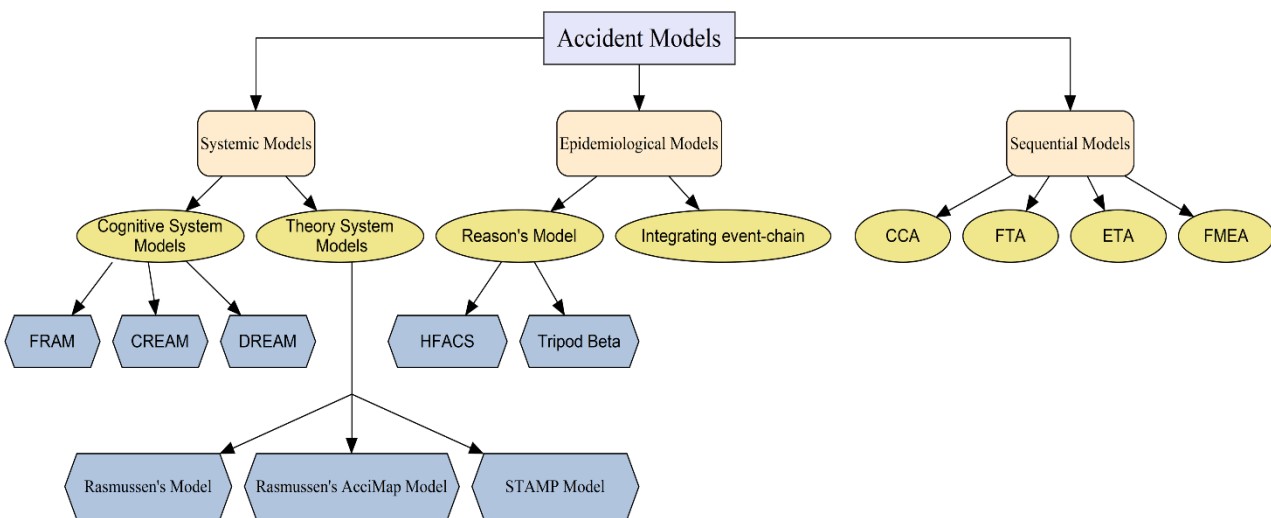

**Figure 1.** Accident model classifications and subset methods [29,36,37].

### 1.3. Epidemiological Accident Models

In these models, accidents are considered to be caused as a combination of "latent" factors such as management functions and organizational culture, as well as "active failures" [50]. Reason's Swiss cheese model is one of the subsets of this category which regards the critical role of organizational safety and the contributory factors of failures of the relevant protective barriers. In this model, the human errors that directly interact with the regulation of the process or technology are the first leading factors for inducing the accidents [51]. In the Reason's model, the dynamics of the accident causation states that failures are transient between barriers, and holes (latent errors) are moving continuously [52]. Bow-tie [53], Threat and Error Management [54], and Tripod [55] are other examples of the models in this category where the use of protective barriers compatible with probable failures is common. The epidemiological models are static and follow the causal pattern in sequential models. Therefore, it may be difficult to also find the explicit factors or critical causes [48,56]. In contrast, the interactions among organizational factors which lead to accidents in the sociotechnical system are more complex and dynamic than the sequential and epidemiological models [57].

### 1.4. Systemic Accident Models

The causes of new accidents in complex sociotechnical systems do not necessarily result from simple defects, and leading factors for accidents occurring in such systems are relevant to the interactions among the system components [21,58,59]. According to the sociotechnical theory, since human and social identities are integral parts of the technical systems, an organization can fulfil its objectives by optimizing the technical as well as the social aspects of the system rather than by merely optimizing the technical aspects of the system [60–62]. Therefore, in order to investigate the causes of accidents in sociotechnical systems, it is necessary to understand the interactions among the principal aspects (e.g., social, technical, human, and organizational) of the system.

Modern sociotechnical systems have drastically modified human activities over the past decades. One of the most noticeable examples of such a shift is the transition from predominantly manual tasks to more cognitive and knowledge-based ones. In fact, various failures and safety issues have already emerged and most of the accidents in such systems cannot be analyzed sufficiently using traditional accident models. Therefore, a new model for risk and safety management with the basis of systems theory was also introduced as a systemic accident model [48].

In systemic models, the study of accidents is based on the uncommon interrelationship and unusual conditions related to accidents. This indicates that there is variability in the system and in order to prevent uncontrollable variability, which is intolerable for the system and leads to an accident, the system performance should be monitored continuously [63]. Some notable systems-modeling approaches of this type include STAMP [39], AcciMap, the hierarchical sociotechnical framework [64] and FRAM [48]. Theoretically, these models are similar; however, their development, methodology, and outputs might differ considerably. These models are described further down.

1.4.1. Rasmussen's Sociotechnical Framework and AcciMap Accident Analysis Technique Overview

The concept of Rasmussen's framework for risk management is based on the control theory, in which the control of system processes is a main concern of safety. In other words, in this framework view, accidents in the sociotechnical systems result from a control problem. Rasmussen's structure of risk management in the sociotechnical systems consists of several levels, from the legislator to the operator (top-down) of the system, respectively (Figure 2). This framework is the basis for the AcciMap accident analysis model [64,65]. Accordingly, the main approach in the AcciMap is the analysis of causal chains of events in the selected accident scenarios using a cause-consequence chart with the aim of analyzing the control layers of the sociotechnical system at the lowest level. On the other hand, in order to extend the cause-consequence chart, a vertical analysis of the mapped accident contributing factors at the hierarchical levels must be conducted [66].

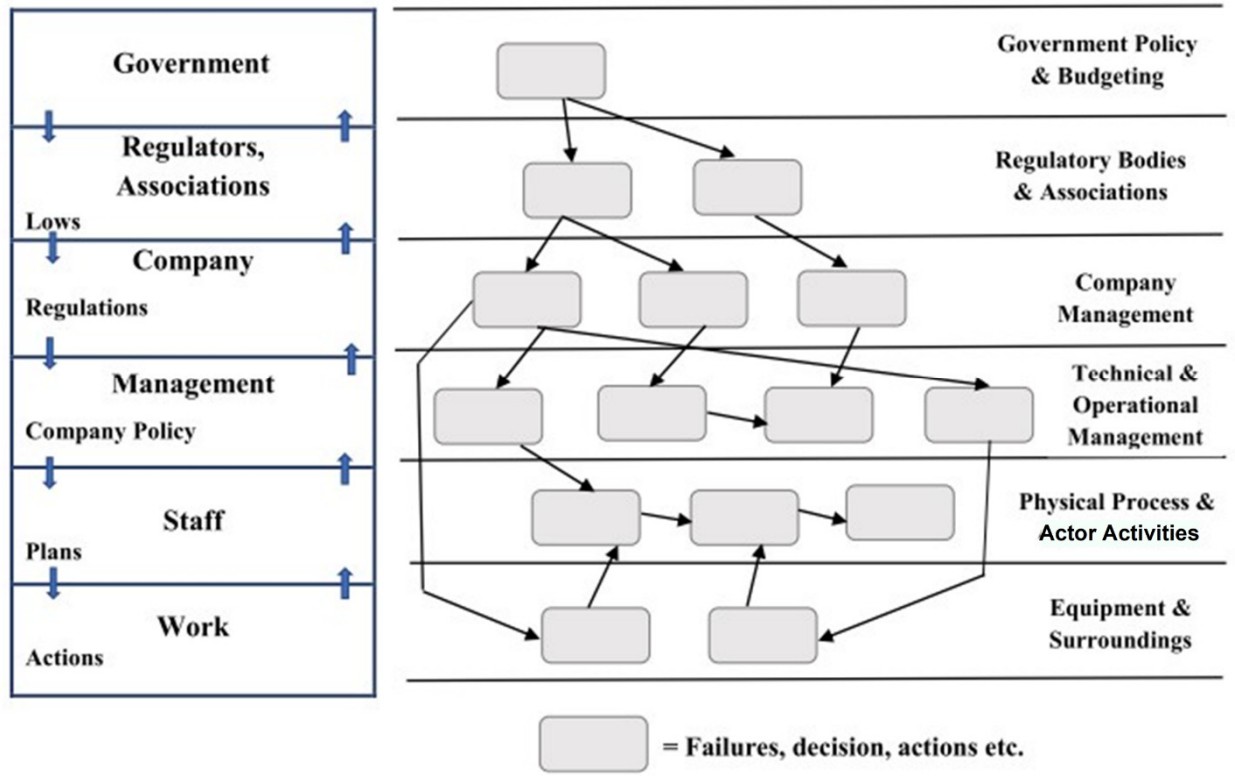

**Figure 2.** Rasmussen's Framework and AcciMap technique [65].

1.4.2. STAMP Analysis Approach Overview

STAMP is a new non-linear system-based accident theory established by Leveson (2011). According to this model, system components are interrelated and enforced by the specific safety constraints [42]. This theory allows for the determination of the dynamics of the interrelationships between system components, as well as a better description of the systems' degree of complexity and technical originality [42].

From the perspective of STAMP, the system is described as a control structure that includes control and feedback loops, and the superior level controls the lower level by applying safety restrictions. Controls and feedbacks are transmitted through every control loop via a collection of relative channels (Figure 3). In the view of organization, controls can be over economic practices and priorities, as well as feedback on reportages and requisitions [42]. Accidents, according to STAMP, are caused by inadequate system components controls which contribute to unsafe component interactions and failures [28].

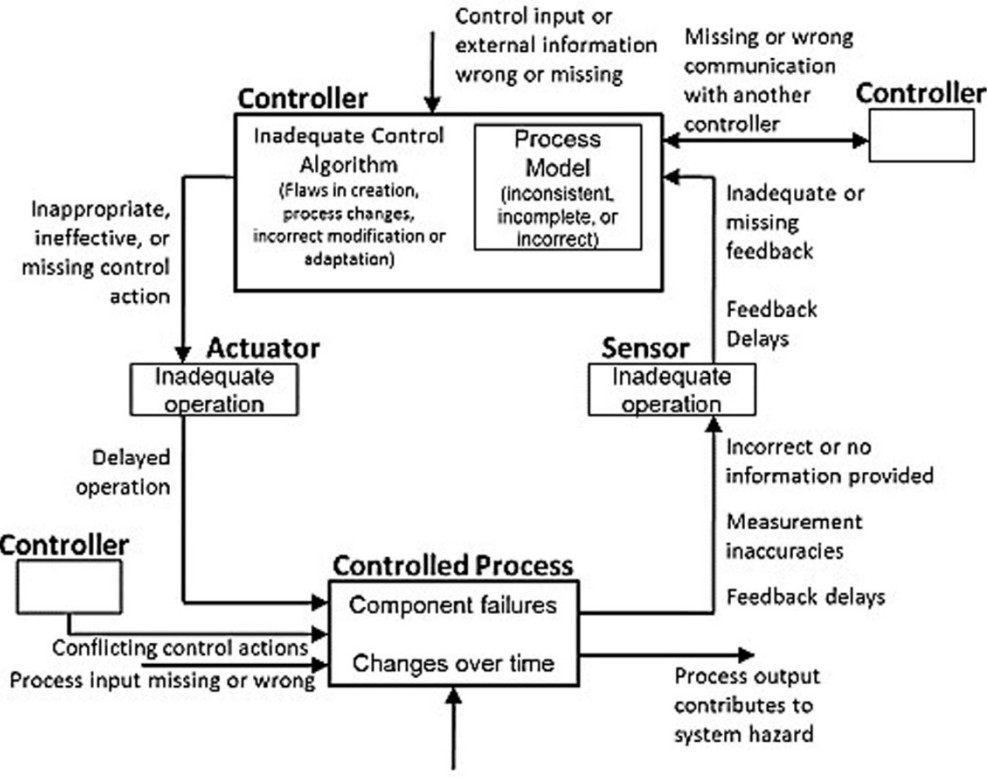

**Figure 3.** General factors in unsafe control used to create STAMP model [42].

STPA and CAST are the two methodologies to be extended and developed from the general STAMP theory. These techniques are usually employed in the analysis of hazards and accidents, respectively [42].

1.4.3. FRAM Analysis Approach Overview

FRAM was first presented as a tool for analyzing accidents in complex systems—particularly, with the aim of evaluating how the functions of a system can interact and trigger accidents. The term "function" refers to the tasks, activities, or components that a system performs or employs in order to achieve a goal [67]. FRAM enables the analysis of the complicated non-linear relationships among functional activities. It also allows for evaluation of the way that functions interact to induce an accident [48]. FRAM can also be utilized for accident analysis and risk assessment based on the operational perspective and the unpredictability of functions [68].

This model has been applied in accident analysis to determine the cause of the accident by documenting typical system performances and their variability in order to manage them. When the method is used with the aim of risk assessment, it examines how variability in one function can affect the performance of other related functions, detects the disruptive variability and finally, controls and minimizes risk levels [69]. Figure 4 demonstrates a schematic of FRAM [48,67] in which each system function is represented by a hexagonal shape with six aspects, representing I as an input, O as an output, P as a precondition, R as a resource, T as a time, and C as a control. Analyzing system performance to develop

models and conceptualize the variability and resonance according to FRAM approach can also be performed using the computer-based tool 'FMV'; http://functionalresonance.com/ 14 June 2021) [70].

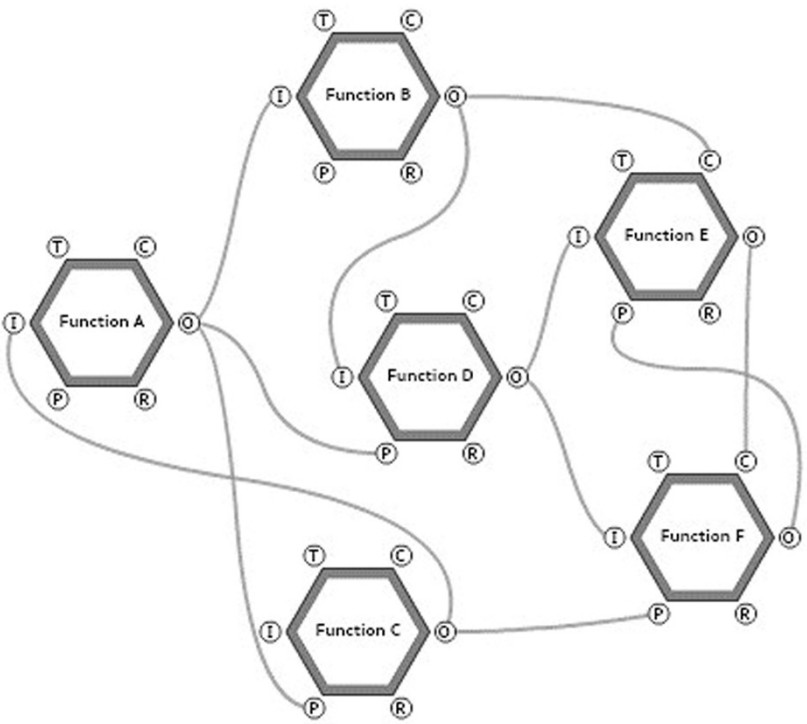

**Figure 4.** General model of FRAM [71].

## 2. Materials and Methods

### 2.1. Search Strategy

We began this investigation by formulating the title as a query in order to locate all papers published in this context. The following question, 'how many articles have been published describing the application of systemic accident analysis models (AcciMap, STAMP, and FRAM)?' was then taken into account and according to the lines of our search, several selected keywords and limiters were used as well: ("STAMP" OR "CAST" OR "STPA" OR "FRAM" OR "AcciMap" OR "Rasmussen's risk management framework" OR "Rasmussen's framework" OR "systemic accident models") AND ("accident analysis" OR "risk assessment" OR "hazard analysis"). Published studies from five international databases (Scopus, Medline/PubMed, Web of science, Science Direct and Google Scholar) were searched. When scanning databases, our search was limited to articles published in the English language with publication dates from 1 January 1990 to 1 October 2021.

### 2.2. Research Screening and Eligibility Criteria

In order to select the studies for inclusion in the current systematic review, we used the Preferred Reporting Items for Systematic Reviews and Meta-analysis (PRISMA) methodology. In the identification phase of this method, after downloading the relevant studies, the duplicates, non-English language research, review articles, letters, and conference proceedings were excluded from our list. Following that, the titles and abstracts of the papers were examined in order to identify those that were particularly relevant. For more screening, full text articles were then retrieved.

The eligibility of the selected papers was then assessed according to predefined inclusion and exclusion criteria.

The inclusion criteria were as follows: original articles that used the AcciMap, STAMP, and FRAM methodologies in their analyses, studies conducting a systemic analysis with the

goals of improving the system safety and resilience through system redesign, and articles that combined other accident analysis methods with systemic methods.

Studies were excluded if they had different data sources, study dates and used additional analyses with either incomplete or insufficient coverage of the systemic models in their methodologies.

In cases where it was not possible to select suitable papers according to the defined criteria, we studied the full text of the paper and if appropriate, it was selected. Finally, we reviewed the full text of the selected articles and extracted information and included them in the tables with the relevant titles.

## 3. Results

### 3.1. Descriptive Results

According to the study plan, 527 records were collected, as shown in Figure 1. Prior to performing screening, 125 duplicates and non-English papers, along with four letters and conference proceedings were excluded from the first list. The anthology of results was then reduced to 398. It should also be noted that this study focused on the research literature that were consistent with our methodology, study goals, and method of application. Additionally, papers that combined alternative methodologies with systemic models to improve their findings were considered. We excluded 167 studies after an examination of the remaining abstracts in terms of relevance. A more thorough analysis of the selected publications' methods and results sections resulted in the elimination of a further 64 papers. Eventually, 63 papers were selected for conducting the analyses in the current study. The results of the search are depicted in the PRISMA flow diagram (Figure 5). Furthermore, as shown in Figure 6, the frequency of 63 systemic methods studies were presented. Accordingly, among 25 AcciMap studies, seven papers were published from the years 2003 to 2010 and 18 works were published from 2011 to 2021. This frequency for 16 STAMP studies in similar ranges was 1 and 16 with a higher frequency in 2018. For 22 FRAM studies, the frequency was 1 and 21, with a higher frequency in 2021. Overall, considering the trend of using these methods, the number of articles increased from 2016, which indicated their capability to understand the behaviors of complex sociotechnical systems.

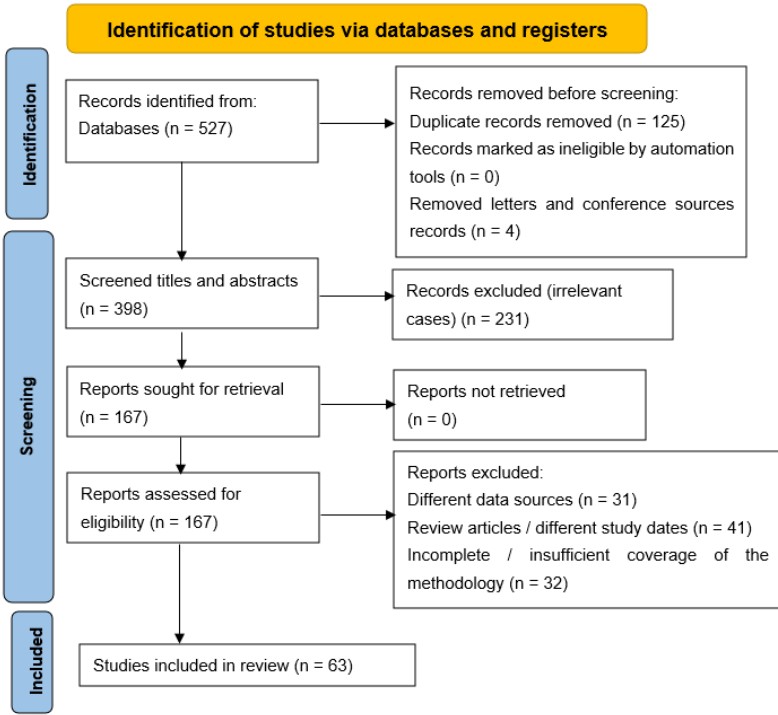

**Figure 5.** PRISMA 2020 flow diagram of the structured literature review.

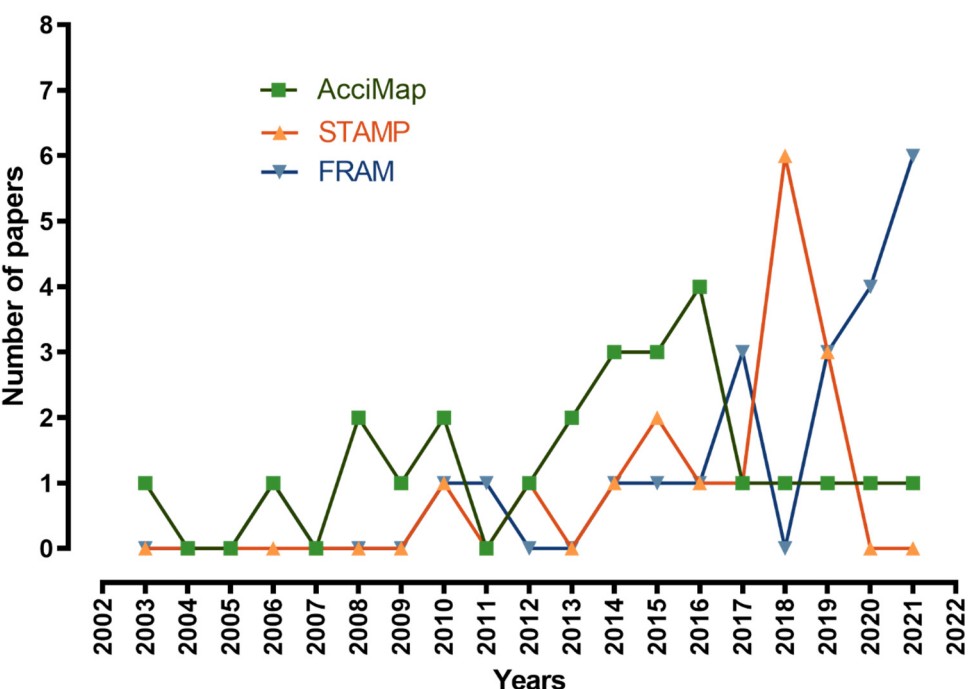

**Figure 6.** The frequency of use of per techniques in recent 20 years by researchers.

*3.2. Key Findings of AcciMap Studies*

As a result of searching the aforementioned databases, we found 25 publications that employed the AcciMap approach to analyze an incident or accident and conduct a safety or risk assessment. Of the AcciMap investigations, 44% (11 studies) and 24% (6 studies) were, respectively, undertaken in the transportation and public health sectors.

Two of the six studies found in the context of healthcare systems had considered the complex interactions among all levels of a complex sociotechnical system using the logic gates or decision trees incorporated with AcciMap. This was to particularly demonstrate the priority and sequence of determined causality for designing public policies by reducing the risk levels in complex systems and investigating the disasters and outbreaks related to the water distribution systems in Canada.

They found a distinction between low-level physical and individual variables, and similar causes of events at the governmental and regulatory factors level [72,73].

Two additional studies conducted in the United Kingdom assessed the level of safety and examined the major events and factors contributing to outbreaks in the food production industry in order to proactively prevent accidents and improve the safety management system [74,75].

One study within the scope of public health examined the factors that contribute to infection outbreaks and provided strategies and interventions for limiting and preventing their occurrence [76]. Additionally, four studies employed AcciMap to connect risk management, accident analysis, and learning from accidents in the context of outdoor recreation. Two studies, on the other hand, utilized a hybrid method to better support the implementation of the AcciMap technique. One of these studies used AcciMap in combination with the CWA to identify accident-related variables and describe conditions within which the accidents occurred. CWA also specified constraints that affect system behavior [77]. Another study used AcciMap in conjunction with the fuzzy ISM and Matrix of Cross Impact Multiplications in which fuzzy ISM was used with the aim of determining the interactions and the hierarchical representation of contributing factors, and Matrix of Cross Impact Multiplications was implied for categorizing and determining the most important factors [78]. Furthermore, two other studies utilized a coding template for the AcciMap technique to quantitatively assess the relationships among accident causes based on the reported frequency of incidents [43,75]. In this regard, another paper was allocated

the codes to accident contributing factors to create a contextual view of the event. They demonstrated the time and place in which decisions and responses were performed [79]. Akyuz et al. applied ANP methods to determine the priority of accident related factors via weighting factors [80]. Other publications were performed in the contexts such as marine, disaster response, navigation, civil engineering, systems thinking principles, and healthcare-related incidents. Overall, AcciMap was used in studies with six hierarchical levels developed based on Rasmussen's (1997) framework. A few works used the five levels of AcciMap and one depicted the contributing factors in the outcome level [81–83]. Table A1 outlines the details of these works (Appendix A).

### 3.3. Key Findings of STAMP Studies

STAMP was found to be the subject of 16 studies, which are listed in Table A2.

These studies were carried out in a variety of contexts and with multiple objectives. Three of the reviewed studies employed this methodology for the risk assessment and identified abnormal system behaviors and potentially unsafe situations in terms of STAMP-STPA. The results from the risk analysis were also utilized to improve and update situational awareness and to prevent accidents through the introduction of safety limitations [84–87]. Moreover, with the aim of accident analysis, some studies used another form of STAMP (CAST methodology) to model and investigate the control deficiency, flaws or missings in a similar way, based on Leveson's (2004) taxonomy, and suggesting corresponding adjustments to increase system sustainability [84–86,88–93].

Additionally, some studies also utilized STAMP in conjunction with other approaches to extend their research beyond the control flaws to fundamental patterns of failures and their implications for the organization's compliance and direction of functions [88,89,91,94]. For example, Lower et al. used HFACS combined with STAMP to improve the accident analysis. This framework incorporated the HFACS levels into a controlling structure of STAMP which can depict the interrelationship between human, technical, and the environmental factors and can be used for hazard, safety and accident analysis [95]. Another study used STAMP in conjunction with SD to provide an integrated framework for analyzing and elaborating on the dynamics and interconnections of human error [86]. Generally, it is clear from reviewing studies that they analyzed and investigated the existing components of system structure(s) and did not elaborate on designing systems by relying on safety properties and system resilience. Table A2 provides a summary of the studies (Appendix B).

### 3.4. Key Findings of FRAM Studies

The FRAM model was utilized in 22 studies and in terms of the contexts, aviation accounted for 28% (6 studies) of the total reviewed papers. The model was also used in other contexts such as the construction and transportation industries, hydrocarbon release accidents, public health and chemical process, and hazard and resilience analysis for complex sociotechnical systems and emergency response systems. Risk analysis, accident analysis, comparison with other approaches, and hybrid usage of FRAM combined with other methods were among the main objectives of the papers that employed FRAM methodology. According to Table A3, 16 studies were conducted with the objective of conducting prospective analyses of risk, hazard, safety, and system behavior as a result of complex interactions between sociotechnical system components. Additionally, they provided controlling strategies for minimizing the risk of function variability or functional resonance in order to improve system operation resilience and sustainability.

A group of researchers considered integrating FRAM with other methodologies such as MCs, GMTA, fuzzy logic, and BN to conduct quantitative and more accurate analyses for increasing the methods' applicability [94,96–100]. For instance, MCs was applied for the quantification of performance variability and the determination of critical couplings through allocating score and probability distribution to each variability [96]. In addition, the hybrid framework including TASM and the combination of FRAM and GMTA was applied in aviation settings to provide the concept maps [101]. In another research, Q-FRAM

provided quantitative concepts in which key indicators of performances were excluded from FRAM and allocated to four concepts of resilience, including anticipate, response, monitor and learn via an MSDM hierarchical approach [97]. Fuzzy logic was also used by Slim et al., in which the performance couplings were weighted and variability of the performances was evaluated with the aim of an aircraft de-icing simulation [98].

Furthermore, two retrospective studies employed FRAM-AHP to evaluate the accidents by determining the main and important criteria to identify the essential functions and relationships between them. These papers would ultimately offer recommendations for enhancing the system operation sustainability [102,103]. Table A3 summarizes the findings of these investigations (Appendix C).

## 4. Discussion

The primary goals of this work were to provide an overview of the papers that had employed AcciMap, STAMP, and FRAM methodologies in their analyses—particularly, in order to: identify the major research flows in terms of the accident analysis, risk assessment and safety analysis of sociotechnical systems; to examine the applicability of hybrid methods for modeling the behavior of accidents and sociotechnical systems; to highlight the advantages and disadvantages of these approaches; to describe safety and accident models in terms of safety-I and safety-II as well as safety-III; and to investigate the impact of using system models for enhancing the systems' sustainability.

### 4.1. The Main Research Flows on Three Systemic Approaches

#### 4.1.1. AcciMap Approach

According to the findings of the related studies, the advantages of the AcciMap application for accident analysis are its ease of use, capability of recognizing factors related to sociotechnical systems, and time-saving nature. Additionally, the most common accident factors at the system's lower levels were "physical practice and operator's function" as well as "instrument and environment". Therefore, it can be concluded that the AcciMap approach in almost all studies can effectively identify the leading factors of the accident, especially at higher levels.

This would also highlight the role of regulatory and governmental bodies in creating a safe environment, demonstrate the interaction of factors at different levels of the system and recommend methods by which the system might be used to prevent accidents proactively [79].

#### 4.1.2. STAMP Approach

The results of related studies showed a similar pattern in which control deficiencies such as "management and the operational process" and the "company" were identified at lower levels of the system.

These contributory factors may be due to the information available to analysts instead of a fixed feature of the accident's leading factors. However, the detected factors at higher levels of the system indicate that controllers at these levels employ strategies to design and provide interventions on human and technical factors which highlight the need for accident prevention.

#### 4.1.3. FRAM Approach

A search of the literature revealed that this method has been used for analysis in construction, transportation, hydrocarbon release accidents, public health, and chemical process sectors. In the FRAM approach, the variability of depicting normal functions is used to determine the emergent behavior of hazards and there is no need for an accident occurrence [103,104]. The model's outputs showed that FRAM has a complicated methodology and procedure and is a challenging model to interpret. As a result, researchers employed novel and innovative techniques to circumvent this problem [99,101,105]. All

reviewed studies which used each of the three mentioned methods also identified multiple contributory factors, functions, and relationships.

### 4.2. Hybrid Use of the Systemic Methods

In this section, we discuss the utilization of systemic techniques integrated with other methods or expanding to the larger methodology as the qualitative and (semi) quantitative approaches. According to the theory of systemic analysis approach, these methods describe and analyze the sociotechnical systems qualitatively. However, a shortcoming is that these methods are only qualitative in nature, particularly due to focusing on constructing a perception model [60]. QRA has shown to have a significant role in effective risk control, as well as addressing the issue of a qualitative structure of systemic analysis methods, mainly in complex sociotechnical systems. Several studies have already proposed quantifying these methods using fuzzy AHP, SME and the MCs and MCMCs methods as the compliment [96,99,106]. The proposed method represents the system more realistically with a quantitative value [100].

MCs allows for reliability indicators to be estimated using real processes and random system behavior simulation in order to make a reality-based scenario by employing a computer-based model. One of the most important applications of MCs is in risk and reliability analysis in the engineering systems. The outputs from MCs simplifies the estimation of the PoFs [107]. According to our literature review, some studies have utilized FRAM and MCs for the enhancement of the traditional safety assessment techniques. For example, Patriarca et al. (2017) used MCs for the first time in their work for quantifying the performance variability in a FRAM model. Their main objective was highlighting the critical functions and links among these functions as well as facilitating the process of safety analysis [96]. Similarly, Kaya et al. integrated MCs as well as a criticality matrix with the FRAM to study how they may be used to enhance the quantification of a system-based risk analysis and critical condition evaluation [94]. Kim et al. proposed a layout to apply the FRAM quantitatively in order to perform the risk assessment. Such layout regarded regulations for variability's aggregation and allocated values for functions and their interactions and therefore showed that the system was more realistic [100]. A FRAM-based tool was also developed utilizing AHP to support in decision-making by quantifying the resilience of urban planning systems [97,99,106].

Contrastingly, Slim et al. engaged predictive FRAM combined with Fuzzy logic to generate numerical indicators for a more comprehensible representation of potential performance variability with the aim of an aircraft system simulation [98]. Moreover, the N-K model was recently introduced by Huang et al. (2021) with the aim of quantitative evaluation of the FRAM model. This model uses the theory of risk pulse according to which the severity of functional coupling can be calculated. According to the model, each coupling with a higher frequency of operation is more likely to have an accident and poses a greater risk. It is worth noting that, unlike earlier studies, this model is constructed on historical data and was not affected by subject matter experts [105]. Furthermore, among AcciMap studies, other authors utilized a coding template for the AcciMap technique to quantitatively assess the accident related factors for assessing the level of safety, proactively preventing the accident and improving the safety management system [43,75]. In order to better support the implementation of this method, AcciMap was also used together with the fuzzy ISM and Matrix of Cross Impact Multiplications to determine and classify the interactions and hierarchical structure of the contributory factors of the accident [78]. Moreover, Wang et al. reported that the simultaneous use of the BN method and systemic methods can provide a quantitative correlation between numerical calculation values and the probability of occurrence [108]. Using the SD method, which explicitly highlights the interrelated time processes, integrated with a BN modeling framework (Dynamic Bayesian Network) for assessing and modeling accidents can overcome the limitations [109]. In this regard, Rong et al. used SD modeling in conjunction with STAMP to demonstrate the dynamic processes which lead to the system changes and to generate safety control structures

with STAMP [86]. Banda et al. also applied the STAMP and BN for the operational use and design of the safety management system [110]. FRAM was also used along with DBN in another study to quantitatively assess and model the system resilience that helps systems to better adjust to unwanted events and restore from major losses. [99].

In the qualitative manner of developing a wider methodology, AcciMap was employed in conjunction with the CWA that enhanced the identification of the causes of accidents and their relationship with the management and system rules in term of the cultural, economic, and social aspects. CWA also specified constraints that affect the system behavior [77]. Kontogiannis et al. investigated the patterns of organizational breakdowns in accidents using the VSM along with STAMP—particularly, with the aim of creating a link between control flaws and organizational breakdowns [85].

However, another study applied Rasmussen's AH combined with FRAM and provided a new structure of FRAM by functional analysis at the hierarchical layers of the system [104]. Additionally, Studic et al. used a hybrid approach including the TASM, the combination of FRAM and GMTA to conduct a system-based modelling of the safety and to provide concept maps in aviation settings [101].

Hence, using the mentioned methods together with systemic accident analysis models as a compliment can improve the process of analysis by providing more reliable information to decision makers. Therefore, future research should consider the dynamic aspects of complex sociotechnical systems in their analysis and more studies should be performed in the context of the resilience analysis of safety management and system behavior using a systemic approach in a dynamic manner.

### 4.3. Advantages and Drawbacks of Systemic Methods

The field of systemic events and analytical modeling describes the system performance and variation control by establishing connections between functions and components of organizational accidents with multiple causes in line with the human factor at different levels of the company in complex modern technologies [111]. They also highlight the influences and possible effects of an unforeseeable occurrence of complex combinations of events and the study of the interactions which exist among system elements. In the present study, we carefully examined the various literature to present the most reasonable and fair presentation of each method and to remain completely neutral in reviewing each method. Moreover, we indicated that each method can be adapted (the mentioned drawbacks will be addressed). According to the peer reviewed studies [39,69,111–113], the main advantages and drawbacks of the three investigated accidents models are shown in Table 1.

It is worth noting that, in accordance with the control characteristics of systemic accident analysis approaches, the application of social, organizational, and managerial controls, collectively referred to as non-technical controls, should be considered in addition to technical controls. As a result, the issue of accident analysis became even more crucial [113] and the primary concern is how inadequate non-technical controls, in addition to the failures of physical controls, can contribute to the occurrence of an accident.

### 4.4. Safety and Accidents Methods in Terms of Safety-I, Safety-II and Safety-III

"Safety" is commonly defined as the absence of an accident, or a system's ability to ensure that the number of harmful events is kept to a minimum and acceptable level [114]. In other words, the purpose of applying safety is to protect, maintain, and gain access to significant and valuable objectives. As a result, safety and sustainability are inextricably linked or even synonymous, as when a system is unsafe, it cannot be sustainable, and vice versa [3].

**Table 1.** The main advantages (Yes), and drawbacks (No) of systemic approaches.

| Descriptions | AcciMap | STAMP | FRAM |
|---|---|---|---|
| Description of accidents with a single diagram | Yes | No | Yes |
| Proximal sequence of events and influences | Yes | Yes | Yes |
| Simplicity of identifying the causes of accident | Yes | No | Yes |
| Identification of contributing factors close to or far from the accident | Yes | Yes | Yes |
| Provision of recommendations for the control structure | Yes | Yes | Yes |
| Description of events and actions | Yes | Yes | No |
| Description of components of system | No | Yes | Yes |
| Providing enough information about system structure | No | No | No |
| Focus on operators and functions | No | Yes | Yes |
| Considering the environmental conditions (equipment and surroundings) | Yes | Yes | Yes |
| Identifying singular root causes for accidents | No | No | No |
| Definition of system boundaries | Yes | Yes | No |
| Providing a context to identify system safety improvements | Yes | Yes | Yes |
| Identification of the control and feedback inadequacies | No | Yes | No |
| Empirical data are not required | Yes | Yes | Yes |
| Minimized level of system information is required for analysis | No | No | No |
| Easier to be implemented | Yes | No | No |
| Providing adequate guidance regarding the methodology | Yes | No | Yes |
| Appropriate for use in a variety of contexts | Yes | Yes | Yes |
| Ability to quantify the accident occurrence and yield probabilities | No | No | No |
| Is not affected by analyst bias | No | No | No |
| Easy to disseminate results to non-experts | No | No | No |

From this perspective, the three concepts of safety (i.e., safety-I, II and III) in relation to accident analysis models are discussed in the following. In the traditional safety-engineering paradigm, safety-I implies that as few things as possible should go wrong during the design process [115,116]. As systems become more advanced and sophisticated, it becomes increasingly vital to focus on enhancing safety while also maintaining the performance modifications to an acceptable level [4].

Complex systems, however, present a different set of safety challenges due to their inherent complexities, ambiguities, and potential for conflicts. Contrary to the apparent significance of these challenges, the traditional management of safety has relatively overlooked this issue [116–120]. According to a safety-I perspective, performance variability should be prevented as it is harmful. In the safety-II approach, is it inevitable, but it may also be useful, so it should be monitored and managed. Therefore, safety-I should progress to a safety-II perspective, in which considerable improvements are established, and we can rely on the system's capacity to react to daily performance variations under varied conditions and maintenance of safety [121]. Therefore, the effort is made for systems to respond to or prevent the hazards by providing suitable controls and interfaces.

In addition, the perspective of the risk assessment "to identify causes and contributory factors" in safety-I should become "understanding the conditions in which performance variability occur" in safety-II [122]. Hence, companies were looking for techniques to implement in varied circumstances according to a safety-II perspective. From a safety-II perspective, since the focus is on monitoring and controlling the determined performance variability, traditional methods are not considered to be sufficient. In that regard, approaches such as the FRAM model [123] were established to explain the system's necessary activities, their connectivity, variability, and resonance, as well as to offer strategies for monitoring and dampening the variability that contributes to accidents [124].

More recently, Hollnagel advanced the concept of safety-III, while its properties remained unspecified beyond those of safety-II. According to this system theory, Leveson defines safety-III as "freedom from intolerable losses" [124,125]. Safety-III defines the concept of accident casualty differently by shifting its focus on the inadequacy of hazard controls as well as relying on the system theory. Considering the concept of sustainability, it also refers to the maintenance of the safety constraints and prevention of losses upon

exposure to the control inadequacy, hazards and unexpected events. Safety-III is primarily concerned with engagement in the design of complex systems' safety management structures in which an appropriate safety culture is created, effective information is available, and the structure of safety management is extensively and carefully constructed. Thus, it is critical to design a sustainable system that is achievable using STAMP, or other tools based on the principle of STAMP (e.g., by using STPA and CAST). System theory approaches identify and analyze controls, hazards, unplanned changes, and associated adaptations in order to mitigate the risk and identify emerging hazards [126].

Nevertheless, it is worth noting that safety-III needs to be extended and improved. It would be preferable if a comprehensive method were developed to analyze sociotechnical systems holistically and to improve integration and communication between human factors and technical aspects for engineers during the early stages of the complex design process, as well as to be capable of being used for highly automated system analysis [126].

*4.5. System Thinking and Improvement in Sustainability of Safety Management*

A system is defined as a collection of interrelated elements that are structured to accomplish a specific purpose. Understanding how system components interact and are organized is critical at the system thinking level. Systems thinking was defined as the science of gathering information about the systems' behavior by creating a rising deep awareness of their components [2]. Moreover, in the systems thinking concept, system components and their environmental interactions have the same importance for the system components behavior. This concept also attends to emergent features, regards complexity, and determines feedback loops, hierarchy, and self-organization, as well as discovering the dynamics and their outcomes [127]. Complex systems have dynamic behavior that needs to be sustained in normal operations. They must also deal with the disturbances and variability of their behavior in order to prevent accidents [26]. Depending on the level of existing risk at work, each company has its own unique health and safety management system. In order to prevent degradation of the system, despite proper design and policy, it is necessary to manage and monitor the system continuously [1].

Therefore, the major element for establishing a sustainable safety management system and ensuring the longevity of safe and healthy organizations is planning and engaging a systemic approach to manage and control the risks. However, in order to execute this, the application of effective methodologies, tools and principles is required. Systems thinking concepts and approaches are able to provide awareness about systems and solve complex issues and for this reason it has been used in a numerous type of fields and disciplines 6. To present a thorough overview of scientists' growing awareness of the notion of safety, and to determine how safety has progressed over time, it is essential to approach these concepts via a system thinking perspective. In order to develop an in depth understanding and awareness of the various layers of the system, this perspective recommends opportunities to act in accordance with one's own human level of awareness. Basically, risk and safety management sought to construct socio-technical systems capable of generating events in the desired locations and preventing or omitting undesirable ones. Nowadays, safety science is concerned with increasing the generation of sustainable systems through using proactive rather than reactive approaches to system safety enhancement. Thus, through increasing system and subsystem awareness, systems thinking approaches can create proactiveness. This approach recommended intervening at the root-cause level rather than focusing on observed symptoms and occurrences. Proposed approaches for this purpose are systemic models that can be used for the analysis of a system's resilience. In that regard, STAMP methodology has already been employed to analyze and assess an organization's sustainable performance or the integration of sustainability in an organization—particularly, by incorporating high-hazard and high-functional-requirement scenarios with predictive objectives [26]. Some studies have used this method in different contexts. They identified abnormal system behaviors and potentially unsafe situations that led to the improvement and updating of system awareness, and the prevention of accidents through the intro-

duction of safety limitations [84,85,88]. It was also employed in accidents analysis in a variety of contexts for identifying insufficient system control limitations and suggesting corresponding adjustments to increase system sustainability [88–93,95,128,129].

Accordingly, sustainable safety management can also be assessed and analyzed through FRAM which is a performance-based risk identification method [48]. This model was employed to evaluate the accidents as well as identify the essential functions and relationships between them and ultimately, offered recommendations for increasing the sustainability of system operations [102,103].

## 5. Conclusions

Our research provided a comprehensive review of systemic approaches of accident analysis utilized in the field of safety investigations. According to the inclusion criteria of this study, a total of 63 research publications employed the three systemic analysis methodologies. AcciMap, STAMP and FRAM were included.

Considering our key findings, all the reviewed research that employed one of these three methods discovered multiple contributing elements, functions, and interactions at various system levels. For instance, for the AcciMap and STAMP methods, the majority of contributing elements and controlling flaws were discovered at the system's lower levels.

Furthermore, the FRAM framework demonstrates the normal functions of the sociotechnical system, defines their variability and identifies the out-of-range variability as the leading indicators of the accident. Due to the relative complexity and difficulty in the interpretation of this model, various novel modifications need to be considered. In addition to an investigation of the advantages and drawbacks associated with the systemic methods, the static and qualitative nature of systemic models and the dynamic structure and ethical control of sophisticated systems were investigated. Safety and accidents analysis methods were also described in terms of safety-I, safety-II and safety-III. Furthermore, this research introduced certain approaches that may be employed in conjunction with the three examined models—particularly, to optimize their applications.

Nonetheless, further research is required to elucidate the critical variables underlying selected systems thinking methodologies for accident causation.

**Author Contributions:** Supervision, E.H; project administration, M.D.; conceptualization, E.Z. and O.V.B.; methodology, M.D.; writing—original draft preparation, M.D.; writing—review and editing, E.Z., M.F., and O.V.B.; funding acquisition, E.H. All authors have read and agreed to the published version of the manuscript.

**Funding:** This research was funded by the Isfahan University of Medical Sciences, Isfahan, Iran (Grant number 340011 and ethical number IR.MUI.REC. 1400-018).

**Institutional Review Board Statement:** Not applicable.

**Informed Consent Statement:** Not applicable.

**Data Availability Statement:** Not applicable.

**Acknowledgments:** This article was extracted from the thesis written by Mahdieh Delikhoon, a student of Occupational Health and Safety Engineering.

**Conflicts of Interest:** The authors declare no conflict of interest.

## Abbreviations

The following abbreviations are used in this manuscript:

| | |
|---|---|
| ILO | International Labor Organization |
| GDP | global gross domestic product |
| STAMP | Systems-Theoretic Accident Model and Processes |
| FRAM | Functional Resonance Accident Model |
| CCA | Cause-Consequence Analysis |
| FTA | Fault Tree Analysis |
| ETA | Event Tree Analysis |
| FMEA | Failure Modes and Effect Analysis |
| STPA | System Theoretic Process Analysis |
| CAST | Causal Analysis based on STAMP |
| FMV | FRAM Model Visualizer |
| CWA | Cognitive Work Analysis |
| ISM | Interpretive Structural Modeling |
| VSM | Viable Systems Model |
| HEMS | Helicopter Emergency Medical Service |
| SD | System Dynamics |
| SMD | Soma Mine Disaster |
| SMS | Safety Management System |
| MCs | Monte Carlo simulations |
| GMTA | Goals-Means Task Analysis |
| BN | Bayesian Networks |
| AH | Abstraction Hierarchy |
| TASM | Total Apron Safety Management |
| DBN | Dynamic Bayesian Network |
| QRA | Quantitative Risk Analysis |
| AHP | Analytical Hierarchy Process |
| SME | Subject Matter Experts |
| MCMCs | Markov Chain Monte Carlo simulation |
| PoFs | Probability of Failures |
| MCDM | Multi Criteria Decision Making |

## Appendix A

**Table A1.** General information and findings from 25 AcciMap studies.

| Objective | Scope of the Study | Main Findings | Location | Reference |
|---|---|---|---|---|
| To find the causes of the disasters related to drinking water distribution systems. | Public health | • Implies complex interactions among all levels of a complex sociotechnical system for designing the public policies to reduce risk in complex systems.<br>• There was a distinction between low-level physical and individual variables, as well as a parallelism between high-level governmental and regulatory factors. | Saskatchewan, Canada | [72] |
| Investigation of leading factors of the water transportation system outbreaks. | Public health | • Describes the causes of accidents.<br>• Specifies how to prevent an accident. | Walkerton, Ontario, Canada | [73] |
| Investigation of the incidents/accidents causality of space programme's launch vehicle. | Aerospace | • Provides a broad framework of leading events, particularly at higher levels, indicating the involvement of regulatory and political authorities in accident formation. | São Paulo, Brazil | [128] |

**Table A1.** *Cont.*

| Objective | Scope of the Study | Main Findings | Location | Reference |
|---|---|---|---|---|
| Assessing the food system safety accidents. | Public health | • Identifies methods for preventing accidents caused by similar sources of hazards. | UK | [74] |
| Analysis of the contributory factors for the infection outbreaks. | Public health | • Demonstrates the strategies and interventions that can be taken to limit and prevent the occurrence of the outbreaks. | Maidstone and Tunbridge Wells, UK | [76] |
| Modeling the events leading up to the Stockwell Underground station accident in July 2005 | Public health | • Proposes a dynamic structure for organization in response to the type of operations and obvious events. | London, UK | [79] |
| Evaluating the led outdoor activity domain. | Led outdoor recreation | • AcciMap is a comprehensive approach to the risk management and accidents analysis developed based on the concept of 'learning from the accident'. | Dorset, UK | [129] |
| Comparing the AcciMap, the HFACS and the STAMP methods to analyze the Mangatepopo gorge tragedy. | Led outdoor recreation | • Describes the failures through the six levels of the studied system. | New Zealand | [130] |
| Assessment of organizational factors in aircraft accidents. | Transport (aircraft) | • The causal remoteness that interlinked to the fatal accident increases as we move up the vertical axis from the accident. | Australia | [131] |
| Examining the incident of rail level crossing system. | Transport (rail) | • In addition to the primary cause of the incident, various system-wide factors contribute to the occurrence of an incident. | Victoria, Australia | [132] |
| Assessment of applicability of systemic frameworks for incident data analysis. | Led outdoor recreation | • Capability of framework to classify contributory factors at various levels of the led outdoor activity was confirmed. | New Zealand | [133] |
| Testing applicability of the method for the analysis the risks associated to the studied case. | Disaster response | • Provides more extensive comprehension of the performance of the case. | Victoria, Australia | [134] |
| Accident analysis using AcciMap, STAMP and SCM methods. | Transport (rail) | • Levels 4 and 5 had the most effective factors in accident and Level 1 of the system, i.e., national government did not include any factors. | Cumbria, UK | [135] |
| Using AcciMap and Analytical Network Process for the assessment of the contributory factors of the marine accidents. | Navigation | • Reveals the main leading factors of accident. <br> • Essential precautionary measures have already been proposed. | Turkey | [80] |
| Identifying the factors that contribute to the collapse of a bridge. | Civil engineering | • Several levels of failure modes were detected. <br> • Demonstrated that human error is a leading contributor element in the occurrence of accidents. | China | [136] |
| Developing a coding template to quantitatively analyze the causes of road freight crashes. | Transport; (road accidents) | • Highlighted the role of systemic approach in enhancement of the safety knowledge. <br> • Recommended preventive measures in the critical domain. | Australia | [75] |
| Identifying the human and systemic causes of outbreaks in the food production domain. | Public health | • The contributory macro and micro factors and their interactions were identified. | South Wales, UK | [81] |
| Using AcciMap and CWA approaches to systemic analysis of a case. | Transport (off-road) | • Hybrid method enhanced the identifying the causes of accidents and their relationship with the management and system rules in term of the cultural, economic, and social aspects. | Queensland; Australia | [77] |

**Table A1.** *Cont.*

| Objective | Scope of the Study | Main Findings | Location | Reference |
|---|---|---|---|---|
| Systemic analysis of South Korea Sewol ferry accident. | Maritime | • Highlighted the importance of allocating resources to safety management in a proactive manner, ongoing monitoring, and having independent and well-informed personnel in charge of continuously monitoring risk to prevent safety migration. | South Korea | [82] |
| Investigating the tragic Sewol Ferry accident. | Maritime; Ferry accidents | • Emphasized the significance of organizational and human variables in the occurrence of accidents. | South Korea | [83] |
| Developing the incidents reporting system as well as emphasizing the importance of learning from the accidents. | Led outdoor recreation | • Indicate the ability of Rasmussen's method of expansion through the safety critical domains. | Australia | [43] |
| Assessing the factors for systemic accidents causation. | Ship grounding accidents | • Used the fuzzy Interpretive Structural Modeling, and Matrix of Cross Impact Multiplications to overcome the limitations of the present AcciMap technique. | China | [78] |
| Performing the risk management proactively. | Road accidents | • Demonstrated that the effectiveness of good management and concern for safety at various levels of the sociotechnical system is a key issue for managing the risks proactively. | Bangladesh | [137] |
| Recognizing the principles of systems thinking in a range of varied systems and events. | Systems thinking tenets | • Declared that the systems thinking tenets can be related to accident causation. | Australia | [138] |
| Evaluating the formalized AcciMap for assessing the causation of accidents. | Healthcare accidents | • Applied leading factors for formulation of safety recommendations. | Scotland, UK | [139] |

## Appendix B

**Table A2.** General information and findings from 16 STAMP studies.

| Objective | Scope of Study | Main Findings | Location | Reference |
|---|---|---|---|---|
| Analyzing the railway accidents and providing improvement measures | Transport (accident in railway) | • Spread accidents analysis in wide sense.<br>• Made impressive urgent actions for case of the study. | China | [84] |
| Using joint STAMP–VSM framework to systemic accidents analysis. | Aviation (HEMS) | • Analyzed the control flaws.<br>• Reviewed the infrastructure of safety.<br>• Models loops and constraints information.<br>• Regarded the conformity and direction of organizational activities.<br>• Developed vast strength interventions | Greece | [85] |
| Demonstration of practicality and validity of the STAMP model. | Industry (a case study in the oil and gas) | • Violations against existing safety constraints that lead to accidents at any level of the organization were identified. | USA | [88] |
| Development of human error causal analysis framework through the STAMP-SD based analysis. | Military | • In whole, 41 leading items related to a broad view of sociotechnical systems were identified and categorized into four types of human errors. | USA | [86] |

**Table A2.** *Cont.*

| Objective | Scope of Study | Main Findings | Location | Reference |
|---|---|---|---|---|
| Demonstration of adaptive and integrated safety management based on STAMP concept. | Maritime Transport System | • The authors recommended using the control loop of STAMP as a basis to develop and implement the integrated safety management. | Finland | [87] |
| Analysis of Korean Sewol ferry accident based on STAMP. | Maritime | • The study developed some continuous improvements and corrective actions to prevent occurrences of catastrophic accidents. | South Korea | [89] |
| Evaluation of hazard control measures effectiveness using STAMP. | Maritime, safety management of traffic | • Determined the level of system hazards.<br>• Identified unsafe situations.<br>• Established control measures of maneuvers.<br>• Updated the situational awareness.<br>• Implemented the real-time safety restrictions. | Finland | [126] |
| Investigated the patient safety incident practices. | Public health | • Offered insights to integration of Human factors and Ergonomics into current practice. | UK | [90] |
| The STAMP was used for the SMD analyzing. | Mine accident | • Identified the inadequate system control constraints.<br>• Suggested the related improvements.<br>• Demonstrated the robustness of method for the cases with high degree of uncertainty. | USA | [91] |
| Analyzing the contributing factors of pipeline leakage and explosion accident. | Process industries accident | • Expanded the causal analysis from a systematic perspective.<br>• Illustrated the utility of model to this case. | China | [92] |
| Analyzing the human factors and taxonomy of system. | Accident analysis | • Analyzed the accidents that occurred due to a major mismatch among components. | Poland | [95] |
| Designing maritime safety management systems. | Safety management systems | • A descriptive process of analysis and key performance indicators was provided for designing maritime safety management systems. | Finland | [116] |
| Hazard analysis of Software-Controlled Systems based on STPA. | Software-Controlled Systems | • A new method HCAT-STPA was proposed for analyzing the software control systems hazards. | China | [140] |
| Using of the STAMP and Bayesian Networks to operational use and design of the safety SMS. | Maritime | • Developed maritime SMS auditing processes. | Finland | [110] |
| Application of systemic methods for the analysis of coal mines accidents. | Coal mines accident | • STAMP model was shown to be a comprehensive and systematic technique.<br>• The model characteristics and analysis processes were complex. | China | [127] |
| Identifying the contributing factors of abnormal behaviors of system that cause process malfunctions using STAMP. | Indoor environment safety | • STAMP effectively identified causes of physical process anomalies. | Japan | [93] |

## Appendix C

**Table A3.** General information and findings of 22 FRAM studies.

| Objective | Scope of Study | Main Findings | Location | References |
|---|---|---|---|---|
| Analyzing aircraft accidents induced by automation autopilots. | Aviation | • Predicted the possible hazard occurrence which may result from complex interactions among human, technological and organizational factors. | Japan | [141] |
| Comparing the two methods: STEP and FRAM | Aviation | • FRAM demonstrated the dynamic interactions of sociotechnical systems.<br>• Described non-linear interrelations among the functions.<br>• Determined the conditions, variability and performance resonance of the functions. | Norway | [142] |
| Analyzing an accident related to the ATM system. | Aviation | • Proposed some recommendation on the system operation resilience.<br>• Indicated that a more profound understanding on the system function is need. | Brazil | [102] |
| Hazard analysis of software system using FRAM and System Hazard Analysis. | Airline | • Established a requirements-based methodology. | Australia | [143] |
| Assessing risk in sustainable construction via FRAM methodology. | Construction | • Control strategies were developed to reduce the risk for function variability or functional resonance. | Brazil | [103] |
| Analysis of the hazards attributed to the sociotechnical system. | Maritime | • Determined the occurrence and aggregation of functions variability.<br>• Illustrated the interactions of functions of system.<br>• Determined how safety constraints are violated. | China | [144] |
| Investigating the compatibility of FRAM model and Rasmussen's AH | Transport (railway) | • Provided a new structure of FRAM by functional analysis at hierarchical layers of the system. | UK | [104] |
| Enhancement of the traditional safety assessment based on semi quantitative FRAM and MCs. | Aviation (ATM system) | • Highlighted the critical functions and critical links among these functions.<br>• Facilitated the safety analysis by considering the system response to different operating conditions and different risk conditions. | Los Angeles | [96] |
| Using a hybrid approach as combining FRAM and TASM to system-based modelling of the safety | Ground handling services | • Advocated the benefits of systemic approaches.<br>• Demonstrated the suitability of the TASM framework for hazard and accident analysis. | UK | [101] |
| Risk assessment and modeling the performance interactions for the maintenance of system. | Hydrocarbon Release Accidents | • The event investigated by connecting various activities and risk influencing factors from a functional perspective. | Norway | [145] |

**Table A3.** *Cont.*

| Objective | Scope of Study | Main Findings | Location | References |
|---|---|---|---|---|
| Quantifying the FRAM. | Resilience Quantification | • The model excluded the main leading indexes. <br> • Resilience bases of the FRAM (anticipate, respond, monitor, learn) were demonstrated. <br> • Overall system variability was demonstrated. | Italy | [97] |
| Predictive performance assessment and improvement of a framework through the integration of FRAM and fuzzy logic. | Complex Sociotechnical Systems | • Generated numerical indicators for a more comprehensible representation of potential performance variability. | Canada | [98] |
| Developing a theory of change to support intervention development. | Public health; care safety | • Supported the theory of change to develop a guide for future safety interventions. | UK | [146] |
| To explore how tensions and contradictions are managed by people. | Public health; patient safety | • Highlighted the main areas of performance variability. | UK | [147] |
| Qualitative risk analysis of shipping operations. | Maritime accident | • Determined the variability of events underlying the accident. <br> • Provided suggestions to examine these events. | Turkey | [121] |
| Risk assessment of highlyautomated vehicles using FRAM. | Automated driving | • The risk and safety assessment were performed. <br> • Proposed recommendations for system design. <br> • Required perspectives on work validation were represented. <br> • Suitability of model was evaluated in detail. | Germany | [148] |
| Analyzing human factors and non-technical skills by modeling the performed activities. | Offshore drilling operations | • Underlined the role of human factors and non-technical skills for the productivity and safety of the work in both normal and critical operation situations. | Brazil | [149] |
| Quantitative assessment of resilience through FRAM and DBN | Chemical process systems | • An effective tool for the purpose of the study was provided. | Kazakhstan | [99] |
| Identifying the challenges within the case of the study | Transition process | • It revealed some challenges affecting the transition process. | Canada | [150] |
| Investigating the applicability of quantified systemic method for risk analysis of the case of study using FRAM and MCs. | Tram operating system | • Systemic method determined functional interactions of the system. <br> • Aggregation of variability was determined. <br> • Comprehensive risk analysis of the case of study was performed. | Turkey | [94] |

Table A3. *Cont.*

| Objective | Scope of Study | Main Findings | Location | References |
|---|---|---|---|---|
| Use of quantitative FRAM for risk assessment. | System of COVID-19 pandemic emergency response | • Potential risks and critical conditions were assessed <br> • Highlighted the role of emergency response strategies at the governance scale. | Republic of Korea | [100] |
| To survey the role of resilience engineering in identifying the system requirements. | Software | • New strategies for meeting the requirements of software for complex systems were represented. | Brazil | [151] |

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
