# Peer review of "Systems Thinking Accident Analysis Models: A Systematic Review for Sustainable Safety Management"

_sustainability, doi:10.3390/su14105869_

Round 1

Reviewer 1 Report

The current review paper presents a systematic review of the applications of three systemic accident analysis models: AcciMap, STAMP and FRAM. The paper first describes the three approaches, presents the review methodology and main findings, highlights the strengths and weaknesses of these models.

Systemic approaches to accident analysis refer to mental models that are most adequate in the context of complex socio-technical system. The current accident analysis models would be definitely very useful and relevant in the context of sustainable management. However, it is not fully articulated in the current work. The paper could be improved by adding more focus on sustainability and how the models can be applied in that context.

There are some minor formatting an language issues:

Specify what is Safety I, II, III (lines 117-118)

Figure 5. PRISMA 2020 flow diagram of the structured literature review - the resolution is too low (line 303)

REPETITION: Theoretically, these models are similar; however, their development, methodology, and outputs might differ considerably. These models are described further down. These models have similar theoretical foundations, but differ in development, methodology and conclusions. These models are explained below. (lines 187-191)

Figure 6. The frequency of using of per techniques in recent 20 years by researchers – line 321

This was particularly perform the followings: line 397-398

Author Response

Response to Reviewer #1:

Question 1) Systemic approaches to accident analysis refer to mental models that are most adequate in the context of complex socio-technical system. The current accident analysis models would be definitely very useful and relevant in the context of sustainable management. However, it is not fully articulated in the current work. The paper could be improved by adding more focus on sustainability and how the models can be applied in that context.

Answer: Thank you for your very careful review of our paper, and for the comments. Section 4.4 and 4.5 have been revised based on this comment and some paragraph with more focus on sustainability.

Question 2) There are some minor formatting an language issues:

Answer: English grammar has been revised.

Question 3) Specify what is Safety I, II, III (lines 117-118)

Answer: With respect to your valuable comment, three paradigms of safety has been described in details in (page 3, lines 118-129).

Forth, to describe the safety and accident models in terms of safety-I “as few things as possible go wrong”, and safety-II “as many things as possible go right” as well as safe-ty-III “freedom from unacceptable losses”. To describe these three paradigms of safety in details: in safety-I paradigm, accidents occur due to failures in system and malfunctions of performances according to which safety management is reactive because the response is when some events were happened and contributory factors were identified. In safety-II paradigm, system is adjusted to respond to events and to eliminate the problems before they were happened and its effort is to make functions “go right.”. Based on this concept, safety management is proactive. Safety-III concept represented that inadequacy of hazards control is the main cause of accidents. In this paradigm, safety management does not regard the identification of root. Instead, it investigates the reason of control malfunctions, preventing accidents and system performance auditing.

 These paradigms also have been also discussed in section 4.4 (Page 15, line 596).

Figure 5. PRISMA 2020 flow diagram of the structured literature review - the resolution is too low (line 303)

Answer: Figure 5. has been reproduced with more resolution.

Question 4) REPETITION: Theoretically, these models are similar; however, their development, methodology, and outputs might differ considerably. These models are described further down. These models have similar theoretical foundations, but differ in development, methodology and conclusions. These models are explained below. (Lines 187-191)

Answer: This has been corrected now (Page 5, line 199-201).

Question 5) Figure 6. The frequency of using of per techniques in recent 20 years by researchers – line 321

Answer: This has been corrected now (Page 8, line 325).

Question 6) This was particularly perform the followings: line 397-398

Answer: This has been corrected now (Page 12, line 464-465).

Reviewer 2 Report

This study is a literature review on accident models, which is very interesting and should be published.

  • On page 3, the authors state that “Generally, there are three categories of accident models: sequential, epidemiological, and systemic models”, please provide the references of the classification or the basis of the classification.
  • The authors refer to the Domino theory, but the domino also refers to chain accidents in the process industry. Please refer to the paper (https://doi.org/10.1016/j.ssci.2020.104618) and shortly discuss the differences.
  • It’s better to put the long table in the appendix.

Author Response

Response to reviewer #2:

Question 1) On page 3, the authors state that “Generally, there are three categories of accident models: sequential, epidemiological, and systemic models”, please provide the references of the classification or the basis of the classification.

Answer: It has been referenced with number 47 in line 142.

Question 2) The authors refer to the Domino theory, but the domino also refers to chain accidents in the process industry. Please refer to the paper (https://doi.org/10.1016/j.ssci.2020.104618) and shortly discuss the differences.

Answer: It has been performed with number 48 in line 154.

Question 3) It’s better to put the long table in the appendix.

Answer: It has been performed. Table 1, 2 and 3 have been put to appendix A, B and C, respectively.

Reviewer 3 Report

1. There is a problem with the section number, there is no 1.2 after 1.1; 4.1-4.3 should be 4.1.1-4.1.3 under 4.1.
2. Materials and Methods. Section 2 simply describe the method of selecting literature, and the content is not substantial. Although the preferred reporting Items for Systematic Reviews and Meta-analysis (PRISMA) methodology is mentioned, it is too simple.
3. Line 289, According to the study, should clearly state.
4. Figure 5 is too vague and introduces the process of selecting literature; Figure 6 frequency analysis, please state your opion.
5. 3.2-3.4 introduced the 63 articles in three means. it was unnecessary, and there was no in-depth summary and elaboration.
6. Line 398, the major research flows, it is not clear what the process is, it should be described in detail.
7. Section 4.4 only shows that these methods are systematic methods of qualitative structure, which need to be used in conjunction with QRA. It is only a simple review without substantive research content.
8.Line 479: Advantages and drawbacks may be problematic.
9. Why are the characteristic indicators in Table 4 ? What are the advantages and disadvantages of the three methods reviewed?
10. What was the motivation for writing this review? What's the purpose? There is no comments.
11.English grammar needs to be revised in some places. Line 22: and finally ?
12.Section  4.6 should clarify the relationship between safety-I-II-III and the three methods.
13. Section 4.7 Systematic thinking and improvement of sustainability, it is simple and the improvement is not explained. Line570-571, performance-based or FRAM? 
14. Line586 and Line589 are both involved FRAM . it can be stated in the same section. 
15.4 Discussion ideas are confusing. The application fields and advantages of the three models of AcciMap STAMP FRAM are mentioned more, and the limitations of the model itself or the problems that cannot be solved in several security modes are described less. Table 4. It is involved but not Comprehensive and lack of systematic description, such as: the problem of quantification of three models, AcciMap relies too much on subjective judgment, STAMP relies on control structure developers, and the workload of dealing with complex problems is large.
16. In the safety-III mode, what problems can be solved by applying STAMP? What difficulties still exist and how to reflect sustainable risk management are less described, and there are no relevant conclusions.

Author Response

Response to Reviewer #3:

Question 1) There is a problem with the section number, there is no 1.2 after 1.1; 4.1-4.3 should be 4.1.1-4.1.3 under 4.1.

Answer: Thank you for your very careful review of our paper, and for the comments. Section numbers were corrected in the text.

Question 2) Materials and Methods. Section 2 simply describe the method of selecting literature, and the content is not substantial. Although the preferred reporting Items for Systematic Reviews and Meta-analysis (PRISMA) methodology is mentioned, it is too simple.

Answer: Thank you for your very careful review of our paper, and for the comments. We add two paragraph as “Track Changes” function to Materials and Methods as follows:

In this study, we first formulated the title of the study as a question as follows to find all the studies that have been published in this content. How many articles have been published with the aim of using systemic accident analysis models (AcciMap, STAMP, FRAM)? In the following, according to the lines of our search, several selected keywords were used to search as follows: ( “ STAMP ”  OR  “ CAST ”  OR  “ STPA ”  OR  “ FRAM ”  OR  “AcciMap ” OR  “ Rasmussen's risk management framework ” OR “ Rasmussen's framework ” OR  “ systemic accident models ” )  AND  ( “accident analysis ”  OR  “ risk assessment ” OR  “ hazard analysis ” ).

In cases where it was not possible to select suitable papers according to the defined criteria, we read the full text of the paper and if appropriate, it was selected. Finally, we re-viewed the full text of the selected articles and extracted information and entered them in the tables with the relevant titles.

We also add some sentences in this section which is evident in the text (Pages 7-8, line 292, 293, 296, 307-310).

Question 3) Line 289, According to the study, should clearly state.

Answer: The sentence “According to the study plan, 527 records were collected, as shown in Figure 1. The anthology of results was reduced to 125 after the duplicate and non-English documents were removed” was replaced with “According to the study plan, 527 records were collected, as shown in Figure 1. Prior to perform screening, 125 duplicates and non-English papers, along with four letters and conference proceedings were excluded from the first list. The anthology of results was then reduced to 398.” in the text (Page 8, line 314-317).

Question 4) Figure 5 is too vague and introduces the process of selecting literature; Figure 6 frequency analysis, please state your opion.

Answer: with respect to your valuable comment, Figure 5 has been reproduced and a paragraph has been written about Figure 6 as “Furthermore, As shown in Figure 6, the frequency of 63 three systemic methods studies were presented. Accordingly, among 25 AcciMap researches, seven papers were published from the years 2003 to 2010 and 18 works were published from 2011 until 2021. This frequency for 16 STAMP studies in similar ranges were 1 and 16 with the more frequency in the year of 2018, and for 22 FRAM researches were 1 and 21, with the more frequency in the year of 2021, respectively. Overall, considering to the trend of using these methods, the number of articles were increased from 2016, which indicated their capability for the understanding the behaviors of complex sociotechnical systems. (Page 9, lines 323-332).

Question 5) 3.2-3.4 introduced the 63 articles in three means. it was unnecessary, and there was no in-depth summary and elaboration.

Answer: Thank you for your valuable comment. These sections have been revised as follows:

3.2. Key findings of AcciMap studies

 As a result of searching the aforementioned databases, we found 25 publications that had employed the AcciMap approach to analyze an incident or accident and conduct a safety or risk assessment. 44% (11 studies) and 24% (6 studies) of the AcciMap investigations were respectively undertaken in the transportation and public health sectors.

Two of the six studies found in the context of healthcare systems had considered the complex interactions among all levels of a complex sociotechnical system using the logic gates or decision trees incorporated with AcciMap. This has been particularly done to demonstrate the priority and sequence of determined causality for designing the public policies by reducing the risk levels in complex systems and investigating the disasters and outbreaks related to the water distribution systems in Canada.

 They found a distinction between low-level physical and individual variables, and similar causes of event at the governmental and regulatory factors level. [75, 76].

 Two additional studies conducted in the United Kingdom assessed the level of safety and examined the major events and factors contributed to outbreaks in the food production industry in order to proactively prevent accidents and improve the safety management system [77, 78].

One research within the scope of public health had examined the factors that contribute to infection outbreaks and provided strategies and interventions for limiting and preventing their occurrence. [79]. Additionally, four research had employed AcciMap to connect risk management, accident analysis, and learning from accidents in the context of outdoor recreation. Two research, on the other hand, had utilized a hybrid method to better support the implementation of the AcciMap technique. One of these research used AcciMap in combination with the CWA to identify accident-related variables and describe conditions within which the accidents occurred. CWA also specified constraints that affect system behavior  [80]. another study used AcciMap in conjunction with the fuzzy ISM and Matrix of Cross Impact Multiplications in which fuzzy ISM was used with the aim of determining the interactions  and the hierarchical representation of contributing factors and Matrix of Cross Impact Multiplications was implied for categorizing and determining the most important factors. [81]. Furthermore, two other studies  utilized a coding template for the AcciMap technique to quantitatively assess the relationships among accident causes based on the reported frequency of incidents [43, 78].  In this regard, another paper was allocated the codes to accident contributing factors to create the contextual view of event. They demonstrated the time and place in which decisions and responds were performed. [82]. Akyuz et al. applied ANP methods to determine the priority of accident related factors via weighting factors [83]. Other publications were performed in the contexts such as marine, disaster response, navigation, civil engineering, systems thinking principles, and healthcare-related incidents. Overall, AcciMap was used in studies with six hierarchical levels developed based on Rasmussen’s (1997) framework.  A few works used the five levels of AcciMap and one depicted the contributing factors in the outcome level [84-86]. Table 1 outlines the details of these works (Appendix A).

3.3. Key findings of STAMP studies

STAMP was found to be the subject of 16 studies, which are listed in Table 2.

These studies were carried out in a variety of contexts and with multiple objectives. Three of the reviewed studies had employed this methodology for the risk assessment and identified abnormal system behaviors and potentially unsafe situations in terms of STAMP-STPA. The results from the risk analysis were also utilized to improve and update situational awareness and prevent accidents through the introduction of safety limitations [87-90]. Moreover, with the aim of accident analysis, some studies used another form of STAMP, CAST methodology to model and investigate the control deficiency, flaws or missing in similar way, based on Leveson’s (2004) taxonomy, and suggesting corresponding adjustments to increase system sustainability [87-89, 91-96].

Additionally, some other studies also utilized STAMP in conjunction with other approaches to extend their study beyond the control flaws to fundamental patterns of failures and their implications for the organization's compliance and direction of functions [88, 89, 97, 98]. For example, Lower et al. used HFACS combining with the STAMP to im-prove the accident analysis. This framework was incorporated the HFACS levels into con-trolling structure of STAMP which can depict the interrelationship between human, technical, and the environmental factors and can be used for hazard, safety and accident analysis. [98]. Another study used the STAMP in conjunction with SD to provide an integrated framework for analyzing and elaborating on the dynamics and interconnections of human error. [89]. Generally, it can be resulted from reviewed studies that they formally used for analyzing and investigation of the existing components of system structure(s) and did not elaborate on designing system by relying on safety properties and system resilience.   Table 2 provides a summary of the studies (Appendix B).

3.4. Key findings of FRAM studies

The FRAM model had been utilized in 22 studies; and in terms of the contexts, aviation accounted for 28% (6 studies) of the total reviewed papers. The model had also been used in other contexts such as construction and transportation industries, hydrocarbon re-lease accidents, public health and chemical process, hazard and resilience analysis for complex sociotechnical systems and emergency response system. Risk analysis, accident analysis, comparison with other approaches, and hybrid usage of FRAM combined with other methods were among the main objectives of the papers that employed FRAM methodology. According to table 3, 16 studies were conducted with the objective of conducting prospective analyses of risk, hazard, safety, and system behavior as a result of complex interactions between sociotechnical system components. Additionally, they provided con-trolling strategies for minimizing the risk of function variability or functional resonance in order to improve system operation resilience and sustainability.

A group of researchers has considered integrating FRAM with other methodologies such as MCs, GMTA, fuzzy logic, and BN to conduct quantitative and more accurate analyses for increasing the methods' applicability [99-104]. For instance, MCs was applied for quantification of performances variability and determination of critical couplings through allocating score and probability distribution to each variability [99]. In addition, the hybrid framework including TASM and the combination of FRAM and GMTA was applied in aviation settings to provide the concept maps [105]. In another research Q-FRAM provided quantitative concept in which key indicators of performances were excluded from FRAM and were allocated to four concepts of resilience including anticipate, response, monitor and learn via MSDM hierarchical approach [100]. Fuzzy logic was also used by Slim et al. in which the performance couplings were weighted and variability of performances was evaluated with the aim of an aircraft de-icing simulation [101].

Furthermore, two retrospective studies employed FRAM-AHP to evaluate the accidents by determining the main and important criteria to identify the essential functions and relationships between them. These papers would ultimately offer recommendations for enhancing the system operation sustainability [106, 107]. Table 3 summarizes the findings of these investigations (Appendix C).

Question 6) Line 398, the major research flows, it is not clear what the process is, it should be described in detail.

Answer: Thank you for your valuable comment. The sentence has been revised and described as “to identify the major research flows in terms of accident analysis, risk assessment, safety analysis and the type of sociotechnical systems (page 11, lines 414-415). The research flows of reviewed papers for AcciMap, STAMP and FRAM have been described in 3.2-3.4 sections. (mentioned as the answer of Question 5)

Question 7) Section 4.4 only shows that these methods are systematic methods of qualitative structure, which need to be used in conjunction with QRA. It is only a simple review without substantive research content.

Answer: Thank you for your valuable comment. This section has been revised as follows:

Revised: 4.2. Hybrid use of the systemic methods

In this section, we discuss about the utilization of systemic techniques integrated with other methods or expanding to the larger methodology as the qualitative and (semi) quantitative approaches. According to the theory of systemic analysis approach, these methods describe and analyze the sociotechnical systems qualitatively. However, such shortcoming exists that these methods are only qualitative in nature; particularly due to focusing on constructing a perception model [61]. QRA has shown to have a significant role in effective risk control as well as addressing the issue of qualitative structure of systemic analysis methods mainly in complex sociotechnical systems, several studies have already proposed quantifying these methods using fuzzy AHP, SME and the MCs and MCMCs methods as the compliment [99, 102, 109]. The proposed method represents the system more realistically with quantitative value [104].

MCs allows for reliability indicators to be estimated using real processes and random system behavior simulation in order to make a reality-based scenario by employing a computer-base model. One of the most important applications of MCs is in risk and reliability analysis in the engineering systems. The outputs from MCs simplifies the estimation of the PoFs [110]. According to our literature review, some studies have utilized FRAM and MCs for the enhancement of the traditional safety assessment techniques. For example, Patriarca et al. (2017) for the first time used MCs in their work for quantifying the performance variability in FRAM model. Their main objective was highlighting the critical functions and links among these functions as well as facilitating the process of safety analysis [99]. Similarly, Kaya et al. integrated MCs as well as a criticality matrix with the FRAM to study how they may be used to enhance the quantification of system-based risk analysis and critical condition evaluation [103]. Kim et al. proposed a layout to apply the FRAM quantitatively in order to perform the risk assessment.  Such layout regarded regulations for variability's aggregation and allocated values for functions and their interactions and therefore showed that the system was more realistic [104]. A FRAM-based tool was also developed utilizing AHP to support in decision-making by quantifying the resilience of urban planning systems [100, 102].

Differently, Slim et al. engaged predictive FRAM combined with Fuzzy logic to generate numerical indicators for a more comprehensible representation of potential performance variability with the aim of an aircraft system simulation [101]. Moreover, the N-K model was recently introduced by Huang et al. (2021) with the aim of quantitative evaluation of the FRAM model. This model uses the theory of risk pulse according to which the severity of functional coupling can be calculated. According to the model, each coupling with a higher frequency of operation is more likely to have an accident and poses a greater risk. It's worth noting that, unlike earlier studies; this model is constructed on historical data and was not affected by subject matter experts. [111]. Furthermore, among AcciMap studies, other authors utilized a coding template for the AcciMap technique to quantitatively assess the accident related factors for assessing the level of safety, proactively pre-venting the accident and improving the safety management system [43, 78]. In order to better support the implementation of method, AcciMap was also used together with the fuzzy ISM and Matrix of Cross Impact Multiplications to determine and classify the inter-actions and hierarchical structure of the contributory factors of the accident [81]. Moreover, Wang et al. reported that the simultaneous use of the BN method and systemic methods can provide a quantitative correlation between numerical calculation values and the probability of occurrence [112]. Using SD method, which explicitly highlight the interrelated time processes, integrated with BN modeling framework (Dynamic Bayesian Net-work) for assessing and modeling accident can overcome the limitations [113]. In this regard, Rong et al. used SD modeling in conjunction with STAMP to demonstrate the dynamic processes which lead to the system changes and to generate safety control structure with STAMP [89]. Banda et al. also applied the STAMP and BN to operational use and design of the safety management system [97].  FRAM was also used model along with DBN in another research to quantitatively assess and model the system resilience that helps system to better adjust with unwanted events and restore from major losses.  [102].

In qualitative manner of developing a wider methodology, AcciMap was employed in conjunction with the CWA that enhanced the identifying the causes of accidents and their relation with the management and system rules in term of the cultural, economic, and social aspects. CWA also specified constraints that affect the system behavior [80]. Kontogiannis et al. investigated the patterns of organizational breakdowns in accidents using the VSM along with STAMP, particularly with the aim of creating a link be-tween control flaws and organizational breakdowns [88].

In a different manner, however, another study applied Rasmussen's AH combined with FRAM  and provided a new structure of FRAM by functional analysis at hierarchical layers of the system [108].  Additionally, Studic et al. used a hybrid approach including the TASM, the combination of FRAM and GMTA to system-based modelling of the safety and to provide a concept maps in aviation settings [105].

Hence, using the mentioned methods together with systemic accident analysis models as a compliment can improve the process of analysis particularly by providing more reliable information to decision makers. Therefore, future research should consider the dynamic aspects of complex sociotechnical systems in their analysis and more studies should be performed in the context of resilience analysis of safety management and system behavior using a systemic approach in a dynamic manner.

Question 8) Line 479: Advantages and drawbacks may be problematic.

Answer: with respect to your comment, there might be correct that different understanding introduced by Table 4. However, we carefully examined once again the various literatures to present the most reasonable and fair presentation of each method and keep us totally neutral in reviewing each method. Moreover, we indicated each method can be evolved which the mentioned drawbacks will be addressed.

Question 9) Why are the characteristic indicators in Table 4 ? What are the advantages and disadvantages of the three methods reviewed?

Answer: The table 4 has been revised as follows:

According to your valuable comments in this question and question 15 “Comprehensive and lack of systematic description”, the term “Characteristics” was replaced with the “Descriptions” term. In addition, three descriptions in the table 4 which could be problematic were deleted. They were: “visual description of system hierarchy”, “being superior in terms of text based results” and “being superior in terms of graphical representation” and items such as “Identifying singular root causes for accidents”, “Empirical data is not required”, “Minimized level of  system information is required for analysis”, “Ability to quantify the accident occurrence and yield probabilities”, “Is not affected by analyst bias”, and “Easy to disseminate results to non-experts” were added in the table.

Question 10) What was the motivation for writing this review? What's the purpose? There is no comments.

As the focus of safety sciences is on increasement of sustainability of system in face of unexpected events and to reach proactive safety management that are achievable using system thinking approaches, the motivation of this study was to review on articles which used three system based methods and to investigate these approaches in some terms as: first, an overview of the papers that had applied the methodologies of AcciMap, STAMP, FRAM in their analyses to synthesize the theoretical and experimental findings; particularly for recognizing the main research flows. Second, to examine the application of mentioned approaches combined with other methods for modeling causal factors of the accidents and the behavior of sociotechnical systems. Third, highlighting the advantages and dis-advantages of these approaches to explore the opportunities for research and practice. Forth, to describe the safety and accident models in terms of safety-I and safety-II as well as safety-III; and finally, to investigate the impact of employing the systemic models for enhancing the systems sustainability. 

Question 11) English grammar needs to be revised in some places. Line 22: and finally?

Answer: English grammar revised of whole text has been performed.

Question 12) Section 4.6 should clarify the relationship between safety-I-II-III and the three methods.

Answer: this section has been revised as follows:

4.4. Safety and accidents methods in terms of safety-I, safety-II and safety-III

"Safety" is commonly defined as the absence of an accident, or a system's ability to ensure that the number of harmful events is kept to a minimum and acceptable level [117]. In other words, the purpose of applying safety is to protect, maintain, and gain access to significant and valuable objectives. As a result, safety and sustainability are inextricably linked or even synonymous, as when a system is unsafe, it cannot be sustainable, and vice versa [3].

From this perspective, the three concepts of safety (i.e., safety-I, II and III) in relation to accident analysis models are discussed in the following. In the traditional safety-engineering paradigm, safety-I, implies that as few things as possible should go wrong during the design process [117, 118]. As systems become more advanced and sophisticated, it becomes increasingly vital to focus on enhancing safety while also maintaining the performance modifications to an acceptable level [45]. 

Complex systems, however, present a different set of safety challenges due to their inherent complexities, ambiguities, and potential for conflicts that they entail. Contrary to the apparent significance these challenges, traditional management of safety has relatively overlooked this issue [118-121].  According to safety-I perspective, performance variability should be prevented as it is harmful. In safety-II approach, not only it is inevitable, but also may be useful, so it should be monitored and managed. Therefore, safety-I should progress to a safety-II perspective, in which considerable improvements are established; and we can rely on the system's capacity to react to daily performance variations under varied conditions and maintenance of safety [122].  Therefore, the effort is made for system to respond or prevent the hazards by providing suitable controls and interfaces.

In addition, the perspective of risk assessment promoted from "to identify causes and contributory factors" in safety-I should turn into "understanding the conditions in which performance variability occur" in safety-II [123]. Hence, companies were looking for techniques to implement in varied circumstances according to safety-II perspective. From a safety-II perspective, since the focus is on monitoring and controlling the determined performance variability, traditional methods are not considered to be sufficient. In that regard, approaches such as the FRAM model [150] were established to explain the system's necessary activities, their connectivity, variability, and resonance, as well as to offer strategies for monitoring and dampening the variability that contributes to accidents [124].

More recently, Hollnagel advanced the concept of safety-III, while its properties remained unspecified beyond those of safety-II. According to this system theory, Leveson de-fines safety-III as "freedom from intolerable losses" [124, 125]. Safety-III, defines the concept of accident casualty differently by shifting its focus on the inadequacy of hazard controls as well as relying on the system theory. Considering the concept of sustainability, it also refers to the maintenance of the safety constraints and prevention of losses upon exposure to the control inadequacy, hazards and unexpected events. Safety-III is primarily concerned with engagement in the design of complex systems' safety management structures in which an appropriate safety culture is created, effective information is available, and the structure of safety management is extensively and carefully constructed. Thus, it is critical to design a sustainable system that is achievable through the use of STAMP or other tools based on the principle of STAMP (e.g., by using STPA and CAST). System theory approaches identify and analyze controls, hazards, unplanned changes, and associated adaptations in order to mitigate the risk and identify emerging hazards [126].

Nevertheless, it is worth noting that safety-III needs to be more extended and im-proved. It would be preferable if a comprehensive method were developed to analyze sociotechnical systems holistically and to improve integration and communication between human factors and technical aspects for engineers during the early stages of the complex design process; as well as to be capable of being used for highly automated system analysis [126].

Question 13) Section 4.7 Systematic thinking and improvement of sustainability, it is simple and the improvement is not explained. Line570-571, performance-based or FRAM?

Answer: this section has been revised as follows:

4.5. System thinking and improvement in sustainability of safety management

A system is defined as a collection of interrelated elements that are structured to accomplish a specific purpose. Understanding how system components interact and are organized is critical at the system thinking level. Systems thinking was defined as the science of gathering information about the systems behavior by creating a rising deep awareness of their components [2]. Moreover, in the systems thinking concept, system component and their environmental interactions have the same importance for the system components behavior. This concept also attends to emergent features, regards complexity, determines feedback loops, hierarchy, and self-organization, as well as finds out the dynamics and their outcomes [127]. Complex systems have dynamic behavior that need to be sustained in normal operation and deal with  the disturbances and variability  of  their behavior in order to prevent accidents [26]. Depending on the level of existing risk at work, each company has its own unique health and safety management system. In order to pre-vent degradation of system, despite proper design and policy, it is necessary to manage and monitor the system continuously [1].

Therefore, the major element for establishing a sustainable safety management and ensuring the longevity of safe and healthy organizations is planning and engaging a systemic approach to manage and control the risks.  However, in order to execute that, it is required to apply effective methodologies, tools and principles. Systems thinking concept and approaches are able to provide awareness about system and solve complex issues and for this reason has been used in a numerous type of fields and disciplines 6. To pre-sent a thorough overview of scientists' growing awareness of the notion of safety, and to determine how safety has progressed over time, it is essential to approach these concepts via a system thinking perspective. In order to develop an in depth understanding and awareness of the various layers of the system, this perspective recommends opportunities to act in accordance with one's level of awareness. Basically, risk and safety management sought to construct socio-technical systems capable of generating events in the desired lo-cations and preventing or omitting undesirable ones. Nowadays, safety science is concerned with increasing the generation of sustainable systems through the use of proactive rather than reactive approaches to system safety enhancement. Thus, through increasing system and subsystem awareness, systems thinking approaches can create proactiveness. This approach recommended intervening at the root-cause level rather than focusing on observed symptoms and occurrences. Proposed approaches for this purpose are systemic models that can be used for the analysis of resilience of a system. In that regard, STAMP methodology has already been employed to analyze and assess an organization’s sustainable performance or the integration of sustainability in an organization; particularly, by incorporating high-hazard and high-functional-requirement scenarios with predictive objectives [26]. Some studies have used this method in different contexts. They identified abnormal system behaviors and potentially unsafe situations that led to improve and up-date awareness of system and prevent accidents through the introduction of safety limitations [87, 88, 91]. It also was employed to accidents analysis in a variety of contexts for identifying insufficient system control limitations and suggesting corresponding adjustments to increase system sustainability [92-96, 98, 128, 129].

Accordingly, sustainable safety management can also be assessed and analyzed through FRAM which is performance-based risk identification method [49]. This model was employed to evaluate the accidents as well as identify the essential functions and relationships between them and ultimately offered recommendations for increasing the sustainability of system operation [106, 107].

Question 14) Line586 and Line589 are both involved FRAM . it can be stated in the same section.

Answer: this comment has been performed as below:

 Furthermore, the FRAM framework demonstrates the normal functions of the sociotechnical system, defines their variability and identifies the out of range variability as the leading indicators of the accident. Due to the relative complexity and difficulty in interpretation of this model, various novel modifications are required to be taken into account. (Lines 711-715)

Question 15) 4 Discussion ideas are confusing. The application fields and advantages of the three models of AcciMap STAMP FRAM are mentioned more, and the limitations of the model itself or the problems that cannot be solved in several security modes are described less. Table 4. It is involved but not Comprehensive and lack of systematic description, such as: the problem of quantification of three models, AcciMap relies too much on subjective judgment, STAMP relies on control structure developers, and the workload of dealing with complex problems is large.

Answer: with respect to your valuable comment, to response to this section of this comment as “4 Discussion ideas are confusing.”, the discussion sections (4.2-4.5) have been mainly revised

4.2. Hybrid use of the systemic methods

In this section, we discuss about the utilization of systemic techniques integrated with other methods or expanding to the larger methodology as the qualitative and (semi) quantitative approaches. According to the theory of systemic analysis approach, these methods describe and analyze the sociotechnical systems qualitatively. However, such shortcoming exists that these methods are only qualitative in nature; particularly due to focusing on constructing a perception model [61]. QRA has shown to have a significant role in effective risk control as well as addressing the issue of qualitative structure of systemic analysis methods mainly in complex sociotechnical systems,  several studies have already proposed quantifying these methods using fuzzy AHP, SME and the MCs and MCMCs methods as the compliment [99, 102, 109]. The proposed method represents the system more realistically with quantitative value [104].

MCs allows for reliability indicators to be estimated using real processes and random system behavior simulation in order to make a reality-based scenario by employing a computer-base model. One of the most important applications of MCs is in risk and reliability analysis in the engineering systems. The outputs from MCs simplifies the estimation of the PoFs [110]. According to our literature review, some studies have utilized FRAM and MCs for the enhancement of the traditional safety assessment techniques. For example, Patriarca et al. (2017) for the first time used MCs in their work for quantifying the performance variability in FRAM model. Their main objective was highlighting the critical functions and links among these functions as well as facilitating the process of safety analysis [99]. Similarly, Kaya et al. integrated MCs as well as a criticality matrix with the FRAM to study how they may be used to enhance the quantification of system-based risk analysis and critical condition evaluation [103]. Kim et al. proposed a layout to apply the FRAM quantitatively in order to perform the risk assessment.  Such layout regarded regulations for variability's aggregation and allocated values for functions and their interactions and therefore showed that the system was more realistic [104]. A FRAM-based tool was also developed utilizing AHP to support in decision-making by quantifying the resilience of urban planning systems [100, 102].

Differently, Slim et al. engaged predictive FRAM combined with Fuzzy logic to generate numerical indicators for a more comprehensible representation of potential performance variability with the aim of an aircraft system simulation [101]. Moreover, the N-K model was recently introduced by Huang et al. (2021) with the aim of quantitative evaluation of the FRAM model. This model uses the theory of risk pulse according to which the severity of functional coupling can be calculated. According to the model, each coupling with a higher frequency of operation is more likely to have an accident and poses a greater risk. It's worth noting that, unlike earlier studies; this model is constructed on historical data and was not affected by subject matter experts. [111]. Furthermore, among AcciMap studies, other authors utilized a coding template for the AcciMap technique to quantitatively assess the accident related factors for assessing the level of safety, proactively pre-venting the accident and improving the safety management system [43, 78]. In order to better support the implementation of method, AcciMap was also used together with the fuzzy ISM and Matrix of Cross Impact Multiplications to determine and classify the inter-actions and hierarchical structure of the contributory factors of the accident [81]. Moreover, Wang et al. reported that the simultaneous use of the BN method and systemic methods can provide a quantitative correlation between numerical calculation values and the probability of occurrence [112]. Using SD method, which explicitly highlight the interrelated time processes, integrated with BN modeling framework (Dynamic Bayesian Net-work) for assessing and modeling accident can overcome the limitations [113]. In this regard, Rong et al. used SD modeling in conjunction with STAMP to demonstrate the dynamic processes which lead to the system changes and to generate safety control structure with STAMP [89]. Banda et al. also applied the STAMP and BN to operational use and design of the safety management system [97].  FRAM was also used model along with DBN in another research to quantitatively assess and model the system resilience that helps system to better adjust with unwanted events and restore from major losses.  [102].

In qualitative manner of developing a wider methodology, AcciMap was employed in conjunction with the CWA that enhanced the identifying the causes of accidents and their relation with the management and system rules in term of the cultural, economic, and social aspects. CWA also specified constraints that affect the system behavior [80]. Kontogiannis et al. investigated the patterns of organizational breakdowns in accidents using the VSM along with STAMP, particularly with the aim of creating a link be-tween control flaws and organizational breakdowns [88].

In a different manner, however, another study applied Rasmussen's AH combined with FRAM  and provided a new structure of FRAM by functional analysis at hierarchical layers of the system [108].  Additionally, Studic et al. used a hybrid approach  including the TASM, the combination of FRAM and GMTA to system-based modelling of the safety and to provide a concept maps in aviation settings [105].

Hence, using the mentioned methods together with systemic accident analysis models as a compliment can improve the process of analysis particularly by providing more reliable information to decision makers. Therefore, future research should consider the dynamic aspects of complex sociotechnical systems in their analysis and more studies should be performed in the context of resilience analysis of safety management and system behavior using a systemic approach in a dynamic manner.

4.3. Advantages and drawbacks of systemic methods

The field of systemic events and analytical modeling describes the system performance and variation control by establishing connections between functions and components of organizational accidents with multiple causes in line with the human factor at different levels of company in complex modern technologies [114]. They also highlight the influences and possible effects of unforeseeable occurrence of complex combinations of events and the study of the interactions which exist among system elements. In the pre-sent study, we carefully examined the various literatures to present the most reasonable and fair presentation of each method and keep us totally neutral in reviewing each meth-od. Moreover, we indicated each method can be evolved which the mentioned drawbacks will be addressed. According the peer reviewed studies [39, 72, 114-116], the main ad-vantages and drawbacks of the three investigated accidents models, are shown in Table 4.

Table 4. The main advantages (Yes), and drawbacks (No) of systemic approaches.

Descriptions

AcciMap

STAMP

FRAM

Description of accidents with a single diagram

Yes

No

Yes

Proximal sequence of events and influences

Yes

Yes

Yes

Simplicity of identifying the causes of accident

Yes

No

Yes

Identification of contributing factors close to or far from the accident

Yes

Yes

Yes

Provision of recommendations for the control structure

Yes

Yes

Yes

Description of events and actions

Yes

Yes

No

Description of components of system

No

Yes

Yes

Providing enough information about system structure

No

No

No

Focus on operators and functions

No

Yes

Yes

Considering the environmental conditions (equipment & surroundings)

Yes

Yes

Yes

Identifying singular root causes for accidents

No

No

No

Definition of system boundaries

Yes

Yes

No

Providing a context to identify system safety improvements

Yes

Yes

Yes

Identification of the control and feedback inadequacies

No

Yes

No

Empirical data is not required

Yes

Yes

Yes

 Minimized level of system information is required for analysis

No

No

No

Easier to be implemented

Yes

No

No

Providing adequate guidance regarding the methodology

Yes

No

Yes

Appropriate for use in a variety of contexts

Yes

Yes

Yes

Ability to quantify the accident occurrence and yield probabilities

No

No

No

 Is not affected by analyst bias

No

No

No

Easy to disseminate results to non-experts

No

No

No

It is worth noting that, in accordance with the control characteristics of systemic accident analysis approaches, the application of social, organizational, and managerial controls, collectively referred to as non-technical controls, should be considered in addition to technical controls. As a result, the issue of accident analysis became even more crucial [42]; and, the primary concern is how inadequate non-technical controls, in addition to the failures of physical controls, can contribute to the occurrence of an accident.

4.4. Safety and accidents methods in terms of safety-I, safety-II and safety-III

"Safety" is commonly defined as the absence of an accident, or a system's ability to ensure that the number of harmful events is kept to a minimum and acceptable level [117]. In other words, the purpose of applying safety is to protect, maintain, and gain access to significant and valuable objectives. As a result, safety and sustainability are inextricably linked or even synonymous, as when a system is unsafe, it cannot be sustainable, and vice versa [3].

From this perspective, the three concepts of safety (i.e., safety-I, II and III) in relation to accident analysis models are discussed in the following. In the traditional safety-engineering paradigm, safety-I, implies that as few things as possible should go wrong during the design process [117, 118]. As systems become more advanced and sophisticated, it becomes increasingly vital to focus on enhancing safety while also maintaining the performance modifications to an acceptable level [45]. 

Complex systems, however, present a different set of safety challenges due to their inherent complexities, ambiguities, and potential for conflicts that they entail. Contrary to the apparent significance these challenges, traditional management of safety has relatively overlooked this issue [118-121].  According to safety-I perspective, performance variability should be prevented as it is harmful. In safety-II approach, not only it is inevitable, but also may be useful, so it should be monitored and managed. Therefore, safety-I should progress to a safety-II perspective, in which considerable improvements are established; and we can rely on the system's capacity to react to daily performance variations under varied conditions and maintenance of safety [122].  Therefore, the effort is made for system to respond or prevent the hazards by providing suitable controls and interfaces.

In addition, the perspective of risk assessment promoted from "to identify causes and contributory factors" in safety-I should turn into "understanding the conditions in which performance variability occur" in safety-II [123]. Hence, companies were looking for techniques to implement in varied circumstances according to safety-II perspective. From a safety-II perspective, since the focus is on monitoring and controlling the determined performance variability, traditional methods are not considered to be sufficient. In that regard, approaches such as the FRAM model [150] were established to explain the system's necessary activities, their connectivity, variability, and resonance, as well as to offer strategies for monitoring and dampening the variability that contributes to accidents [124].

More recently, Hollnagel advanced the concept of safety-III, while its properties remained unspecified beyond those of safety-II. According to this system theory, Leveson de-fines safety-III as "freedom from intolerable losses" [124, 125]. Safety-III, defines the concept of accident casualty differently by shifting its focus on the inadequacy of hazard controls as well as relying on the system theory. Considering the concept of sustainability, it also refers to the maintenance of the safety constraints and prevention of losses upon exposure to the control inadequacy, hazards and unexpected events. Safety-III is primarily concerned with engagement in the design of complex systems' safety management structures in which an appropriate safety culture is created, effective information is available, and the structure of safety management is extensively and carefully constructed. Thus, it is critical to design a sustainable system that is achievable through the use of STAMP or other tools based on the principle of STAMP (e.g., by using STPA and CAST). System theory approaches identify and analyze controls, hazards, unplanned changes, and associated adaptations in order to mitigate the risk and identify emerging hazards [126].

Nevertheless, it is worth noting that safety-III needs to be more extended and im-proved. It would be preferable if a comprehensive method were developed to analyze sociotechnical systems holistically and to improve integration and communication between human factors and technical aspects for engineers during the early stages of the complex design process; as well as to be capable of being used for highly automated system analysis [126].

4.5. System thinking and improvement in sustainability of safety management

A system is defined as a collection of interrelated elements that are structured to accomplish a specific purpose. Understanding how system components interact and are organized is critical at the system thinking level. Systems thinking was defined as the science of gathering information about the systems behavior by creating a rising deep awareness of their components [2]. Moreover, in the systems thinking concept, system component and their environmental interactions have the same importance for the system components behavior. This concept also attends to emergent features, regards complexity, determines feedback loops, hierarchy, and self-organization, as well as finds out the dynamics and their outcomes [127]. Complex systems have dynamic behavior  that need to be sustained in normal operation and deal with  the disturbances and variability  of  their behavior in order to prevent accidents [26]. Depending on the level of existing risk at work, each company has its own unique health and safety management system. In order to pre-vent degradation of system, despite proper design and policy, it is necessary to manage and monitor the system continuously [1].

Therefore, the major element for establishing a sustainable safety management and ensuring the longevity of safe and healthy organizations is planning and engaging a systemic approach to manage and control the risks.  However, in order to execute that, it is required to apply effective methodologies, tools and principles. Systems thinking concept and approaches are able to provide awareness about system and solve complex issues and for this reason has been used in a numerous type of fields and disciplines 6. To pre-sent a thorough overview of scientists' growing awareness of the notion of safety, and to determine how safety has progressed over time, it is essential to approach these concepts via a systems thinking perspective.  In order to develop an in depth understanding and awareness of the various layers of the system, this perspective recommends opportunities to act in accordance with one's level of awareness. Basically, risk and safety management sought to construct socio-technical systems capable of generating events in the desired lo-cations and preventing or omitting undesirable ones. Nowadays, safety science is concerned with increasing the generation of sustainable systems through the use of proactive rather than reactive approaches to system safety enhancement. Thus, through increasing system and subsystem awareness, systems thinking approaches can create proactiveness. This approach recommended intervening at the root-cause level rather than focusing on observed symptoms and occurrences. Proposed approaches for this purpose are systemic models that can be used for the analysis of resilience of a system. In that regard, STAMP methodology has already been employed to analyze and assess an organization’s sustainable performance or the integration of sustainability in an organization; particularly, by incorporating high-hazard and high-functional-requirement scenarios with predictive objectives [26]. Some studies have used this method in different contexts. They identified abnormal system behaviors and potentially unsafe situations that led to improve and up-date awareness of system and prevent accidents through the introduction of safety limitations [87, 88, 91]. It also was employed to accidents analysis in a variety of contexts for identifying insufficient system control limitations and suggesting corresponding adjustments to increase system sustainability [92-96, 98, 128, 129].

Accordingly, sustainable safety management  can also be assessed and analyzed through FRAM which is performance-based risk identification method [49]. This model was employed to evaluate the accidents as well as identify the essential functions and relationships between them and ultimately offered recommendations for increasing the sustainability of system operation [106, 107].

Moreover, the table 4 has been revised as follows:

According to your valuable comments in this question “Comprehensive and lack of systematic description”, items such as “Identifying singular root causes for accidents”, “Empirical data is not required”, “Minimized level of  system information is required for analysis”, “Ability to quantify the accident occurrence and yield probabilities”, “Is not affected by analyst bias”, and “Easy to disseminate results to non-experts” were added in the table 4.

Question 16) In the safety-III mode, what problems can be solved by applying STAMP? What difficulties still exist and how to reflect sustainable risk management are less described, and there are no relevant conclusions.

Answer: this comment has been responded as below and also as the answer of question 12 (Page 16, lines 630-650).

More recently, Hollnagel advanced the concept of safety-III, while its properties remained unspecified beyond those of safety-II. According to this system theory, Leveson de-fines safety-III as "freedom from intolerable losses" [124, 125]. Safety-III, defines the concept of accident casualty differently by shifting its focus on the inadequacy of hazard controls as well as relying on the system theory. Considering the concept of sustainability, it also refers to the maintenance of the safety constraints and prevention of losses upon exposure to the control inadequacy, hazards and unexpected events. Safety-III is primarily concerned with engagement in the design of complex systems' safety management structures in which an appropriate safety culture is created, effective information is available, and the structure of safety management is extensively and carefully constructed. Thus, it is critical to design a sustainable system that is achievable through the use of STAMP or other tools based on the principle of STAMP (e.g., by using STPA and CAST). System theory approaches identify and analyze controls, hazards, unplanned changes, and associated adaptations in order to mitigate the risk and identify emerging hazards [126].

Nevertheless, it is worth noting that safety-III needs to be more extended and im-proved. It would be preferable if a comprehensive method were developed to analyze sociotechnical systems holistically and to improve integration and communication between human factors and technical aspects for engineers during the early stages of the complex design process; as well as to be capable of being used for highly automated system analysis [126].

Round 2

Reviewer 3 Report

The new version has a lot of changes. Responses to previous reviews are acceptable.